# Positional Fragility in LLMs: How Offset Effects Reshape Our Understanding of Memorization Risks

Yixuan Xu[1,2]    Antoine Bosselut[2]    Imanol Schlag[1]
[1]ETH AI Center, ETH Zürich
[2]School of Computer and Communication Sciences, EPFL
firstname.lastname@inf.ethz.ch   antoine.bosselut@epfl.ch

⚙Alvorecer721/Positional_Fragility_LLM

## Abstract

Large language models are known to memorize parts of their training data, posing risk of copyright violations. To systematically examine this risk, we pretrain language models (1B/3B/8B) from scratch on 83B tokens, mixing web-scale data with public domain books used to simulate copyrighted content at controlled frequencies at lengths at least ten times longer than prior work. We thereby identified the offset effect, a phenomenon characterized by two key findings: (1) verbatim memorization is most strongly triggered by short prefixes drawn from the beginning of the context window, with memorization decreasing counterintuitively as prefix length increases; and (2) a sharp decline in verbatim recall when prefix begins offset from the initial tokens of the context window. We attribute this to *positional fragility*: models rely disproportionately on the earliest tokens in their context window as retrieval anchors, making them sensitive to even slight shifts. We further observe that when the model fails to retrieve memorized content, it often produces degenerated text. Leveraging these findings, we show that shifting sensitive data deeper into the context window suppresses both extractable memorization and degeneration. Our results suggest that positional offset is a critical and previously overlooked axis for evaluating memorization risks, since prior work implicitly assumed uniformity by probing only from the beginning of documents or training sequences.

## 1  Introduction

Large language models (LLMs) have demonstrated the capacity to reproduce substantial portions of their training data verbatim, raising serious concerns about copyright violations [Chang et al., 2023, Karamolegkou et al., 2023] and privacy breaches [Huang et al., 2022]. This capability has fueled increasing legal challenges, with high-profile cases, such as the New York Times lawsuit against OpenAI, highlighting the potential misuse of proprietary content [Freeman et al., 2024]. Industry leaders have not denied these risks, and instead called for regulatory exemptions that would allow LLM training on copyright-protected material [Brodkin, 2025]. Against this backdrop, understanding how LLMs memorize and regurgitate legally sensitive content is no longer a niche technical concern, but a central requirement for the responsible and legal development of LLMs [Rosenthal and Veraldi, 2025] and informed policymaking on this subject.

Existing work has begun to study memorization in LLMs, but key gaps remain. First, most mitigation strategies are applied post-training via fine-tuning, decoding constraints, or unlearning methods. However, recent studies show that these approaches can be bypassed [Park et al., 2024, Nasr et al., 2025], indicating that they are better suited as complementary defenses rather than standalone solutions. This highlights the need to address memorization proactively during pretraining, where

39th Conference on Neural Information Processing Systems (NeurIPS 2025).

retention first happens. Second, prior analyses typically focus on short suffixes prompted from the beginning of training sequences, implicitly assuming that memorization behavior is uniform across the context window. This methodological choice overlooks the reality that user inputs may appear at arbitrary offsets in the content window, and that memorization may vary significantly with position.

In this work, we revisit memorization from a legally motivated perspective, focusing on long-form text segments drawn from published books (via Project Gutenberg) to simulate high-risk copyright scenarios. We pretrain LLaMA-style models (1B, 3B, and 8B) from scratch on 83B tokens, combining curated book data with web-scale educational corpora. This setup grants precise control over training dynamics (such as model scale and batch size), data composition (such as exposure frequency and placement of sensitive content), and evaluation configuration (such as prefix length and position within the context window). Compared to the most extensive memorization analysis to date by Hans et al. [2025], which also targets pretraining-time mitigation, our study employs full-context sequences four times longer and evaluates suffixes at least ten times longer. Together, these design choices enable an unprecedented, comprehensive, and fine-grained memorization analysis.

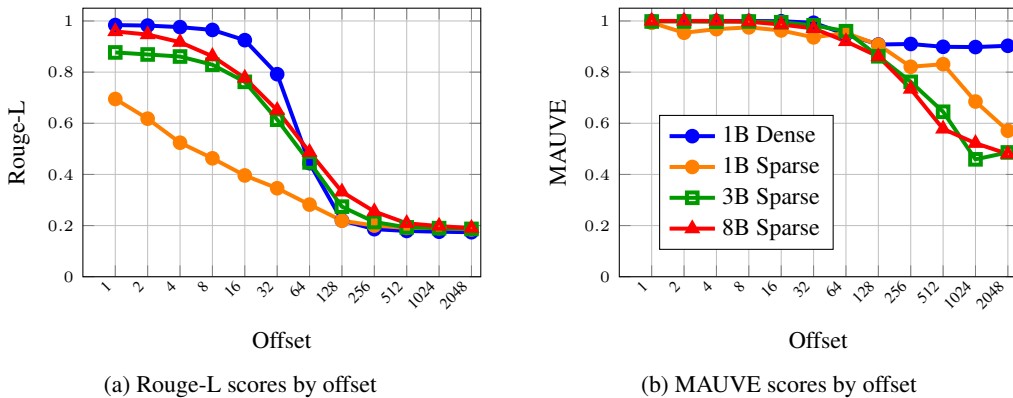

(a) Rouge-L scores by offset        (b) MAUVE scores by offset

Figure 1: Positional fragility in LLMs measured with two complementary metrics: (a) Rouge-L scores quantify verbatim memorization, demonstrating a sharp decline as offset increases; (b) MAUVE scores capture overall language coherence, revealing text degeneration with increasing offset. The dense model results are derived from a 1B model trained exclusively on 10K Gutenberg sequences seen 80 times each, while sparse model results are derived specifically from the frequency-128 bucket of our FM-Probe methodology. All experiments used 500-token prefixes and evaluated on 500-token suffixes. Verbatim memorization and language coherence both degrade sharply as prefix offset increases, except for language coherence in 1B Dense case, revealing positional fragility.

Our main contributions are:

- We identify the **offset effects**: verbatim memorization is most effectively triggered by short prefixes at the beginning of the context window. Memorization recall drops sharply as prefixes are shifted further into the sequence. We attribute offset effects to **positional fragility**: LLMs are highly sensitive to positional shifts as they disproportionately rely on early tokens as retrieval anchors (§ 4).

- We show that memorization failures cause **degeneration**. Under offset or low-frequency conditions, the model produces repetitive and semantically incoherent output, establishing a link between memorization breakdown and text degeneration (§ 5).

- We show that **positional fragility can be leveraged to mitigate memorization:** shifting sensitive sequences deeper into the context window significantly reduces extractability without compromising general performance (§ 7).

- We find that **batch size influences retention:** smaller batches with more frequent updates modestly reduce memorization under identical compute budgets. (§ 8).

- We **reproduce and scale** previous studies in more controlled and legally relevant settings, using longer sequences to better reflect the risks of copyright in the real world (Appendix C).

## 2   Problem Setup

To facilitate the discussion around verbatim memorization we introduce the following nomenclature:

**Text Segment**: A sequence of 8192 tokens from the training corpus, beginning with a BOD token, that fits exactly the context window of the model.

**Prefix** $x$: The input text sequence provided to the model to probe memorization, represented as $x = (x_1, ..., x_n)$ with $n$ tokens extracted consecutively from the training corpus.

**True Suffix** $s$: The text sequence that immediately follows the prefix in the original text segment, represented as $s = (s_1, ..., s_m)$ with $m$ consecutive tokens.

**Generated Suffix** $y$: The text sequence produced by the model given prefix $x$, where $y = (y_1, ..., y_m)$, that matches the length of the ground truth suffix $m$.

**Offset** $o$: The number of tokens to skip in the text segment from the beginning-of-document token BOD before starting prefix extraction.

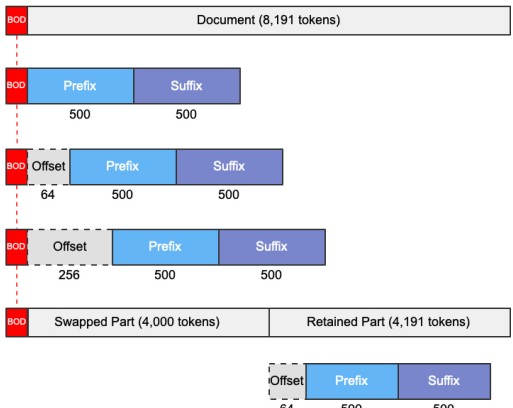

Figure 2: Illustration of prefix-suffix extraction with varying offsets from the document start in Sparse Gutenberg, and from the retained part in Swapped Gutenberg. Not to scale.

**Quantifying Verbatim Memorization**   We follow Nasr et al. [2025]'s definition of verbatim memorization, providing a model with a prefix $x$ from training data and measuring how closely its greedy continuation $y$ matches the true continuation $s$. Prior work predominantly focuses on short prefix probing, using prefixes of size 20 [Carlini et al., 2021], 32 [Biderman et al., 2023, Huang et al., 2024, Duan et al., 2025], 50 [Karamolegkou et al., 2023, Duan et al., 2025, Ippolito et al., 2023], 50–500 [Carlini et al., 2023], and 1998 [Hans et al., 2025]. Suffix lengths are likewise short, typically ranging from 25 to 50 tokens. Nearly all evaluations extract prefixes from the beginning of the document, with only Carlini et al. [2021] sampling prefixes from arbitrary positions, but they only report the highest-recall case. To address these limitations, our evaluation spans prefix lengths from 50 to 5000 tokens, sampled at offsets ranging from 0 to 2048, with suffixes between 50 and 8000 tokens. We report similarity using three metrics that capture increasing levels of verbatim reproduction: Rouge-L (longest common subsequence relative to reference length) [Lin, 2004], LCCS (longest common contiguous substring) [Freeman et al., 2024], and EM (exact match of the entire suffix).

**Measuring Text Degeneration**   We evaluate output quality using three complementary metrics. *Type-token ratio* (TTR) measures lexical diversity as the fraction of unique tokens in the generated suffix [Kettunen, 2014]. *MAUVE* quantifies distributional similarity between model outputs and human-written text [Pillutla et al., 2023]. As a baseline, unrelated suffix pairs yield a ROUGE-L of approximately 0.18. We also report *perplexity* on the reference suffix to assess how well the model predicts the ground-truth continuation. Given a prefix $x = (x_1, \ldots, x_n)$ and its true suffix $s = (s_1, \ldots, s_m)$, perplexity is defined as $\text{PPL}(s \mid x) = \exp\left(-\frac{1}{m} \sum_{t=1}^{m} \log P(s_t \mid x, s_{<t})\right)$. Intuitively, perplexity reflects how many next-token candidates the model finds plausible at each step [Jelinek et al., 2005]. For downstream performance evaluation, we utilize lm-eval-harness [Gao et al., 2024].

## 3   Design

**Data & Model**   We train a decoder-only transformer model following the Llama architecture [Grattafiori et al., 2024], with complete architectural and training details provided in Appendix A.2. Our training corpus combines two complementary sources: (1) Project Gutenberg[1], used to simulate potential copyright infringement via literary texts; and (2) Fineweb-Edu [Penedo et al., 2024], a curated dataset of high-quality educational web content typical in modern LLM pipelines. A 13-gram contamination analysis confirmed a negligible 0.34% overlap between the two datasets, ensuring observed memorization is attributable to our experimental controls.

---

[1]https://huggingface.co/datasets/manu/project_gutenberg

**Dense Gutenberg: Extreme Memorization Study**    Inspired by the setup of Hans et al. [2025], we establish an extreme memorization scenario, where models are trained exclusively on our curated subset of Project Gutenberg text segments comprising 10,000 sequences of 8,192 tokens each. Each epoch constitutes a complete traversal of the entire corpus, with model checkpoints saved at logarithmically spaced intervals (powers of 2) as well as at the final 80th epoch. Since each sequence appears exactly once per epoch, the checkpoint number directly corresponds to the number of exposures to each training sequence, providing a direct measure of how repetition influences memorization behavior.

**Sparse Gutenberg: Realistic Copyright Memorization Simulation**    To create a more realistic scenario, we designed a mixed corpus training configuration where Project Gutenberg excerpts represent just 2% (1.74B tokens) of the training data, with the remaining 98% (81.82B tokens) sourced from FineWeb-Edu. For analyzing how exposure frequency affects memorization, we developed Frequency-Varied Memorization Probe Buckets (FM-Probes), with each bucket containing 500 distinct and randomly sampled Gutenberg sequences that appear at precisely controlled frequencies ranging from 1 to 128 repetitions. By performing a single training pass through this carefully structured corpus, our FM-probe methodology enables concurrent evaluation of memorization patterns across multiple exposure levels without requiring separate training runs.

**Pretraining Factors**    Within the Sparse Gutenberg framework, we systematically investigate several key factors that potentially influence memorization dynamics during pretraining. First, we examine batch size variations (0.5M, 1M, 1.5M and 2M tokens) to assess how optimization granularity affects memorization tendencies. Second, we analyze model scale effects by comparing memorization patterns across architecturally consistent models of 1B, 3B, and 8B parameters. Third, we evaluate how exposure frequency impacts memorization propensity through our FM-Probe methodology.

# 4    Offset: The Missing Dimension in Memorization Evaluation

Despite increasing attention to memorization risks in LLMs, most literature evaluates memorization using prefixes extracted from the beginning of documents [Hans et al., 2025, Carlini et al., 2023, Kiyomaru et al., 2024]. However, memorized content can appear anywhere within a document—not just at the start. At the same time, little attention has been paid to how the absolute position of the prefix within the model's context window—the offset—affects memorization behavior. In this section, we show that both verbatim memorization and language generation quality are highly sensitive to offset, and whether longer prefixes increase verbatim recall is itself offset-dependent.

## 4.1    Shorter Prefixes Outperform Longer Ones in Triggering Memorization

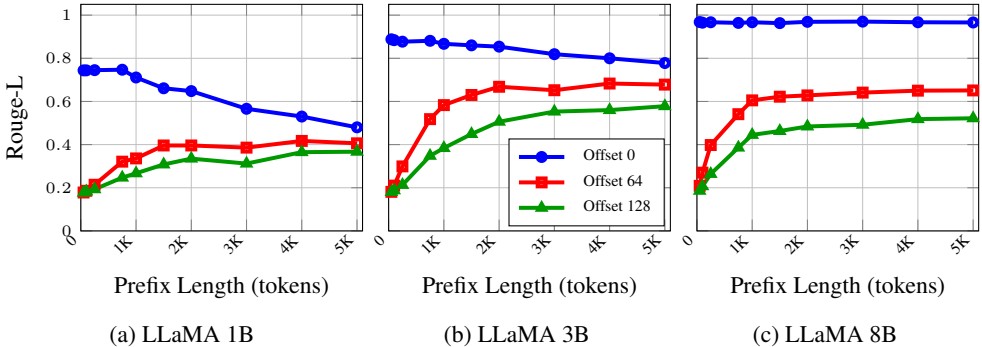

(a) LLaMA 1B          (b) LLaMA 3B          (c) LLaMA 8B

Figure 3: Impact of prefix length on positional fragility across LLaMA models (1B, 3B, 8B), measured by Rouge-L scores on frequency-128 bucket with 500-token suffixes under Sparse Gutenberg setup. With zero offset (blue), smaller models show memorization degradation as prefix length increases, while the 8B model maintains high scores. At non-zero offsets (red, green), all models require longer prefixes to trigger substantial memorization, with the 3B model showing slightly higher memorization than the 8B model at offsets 64 and 128, indicating larger models do not worsen positional fragility.

While previous research suggests that longer prefixes can enhance verbatim recall, we find that this observation is incomplete and potentially misleading. Our results reveal a more efficient condition for triggering memorization: using short prefixes that align with the beginning of the context window during training. As shown in Figure 3, at offset 0, **short prefixes yield the highest recall**, particularly in smaller models, whereas **longer prefixes and longer offsets reduce extractable memorization**. Note that this result does not contradict previous work. As we show in Table 1, with a fixed prefix of 50 tokens, we can induce perfect reproduction of suffixes of up to 8,000 tokens long, especially when the target sequence is frequently encountered during training. However, such an evaluation does not take into account larger prefixes and non-zero offsets.

Table 1: Suffix length and sample counts of **perfect matching** by model size and repetition frequency at offset 0, given a fixed 50-token prefix. Frequencies under 8 showed no verbatim memorization over 50 tokens and are therefore omitted.

| Freq. | LLaMA 1B | | LLaMA 3B | | LLaMA 8B | |
|---|---|---|---|---|---|---|
| | Length | Count | Length | Count | Length | Count |
| 8 | - | 0 | - | 0 | 50 | 2 |
| 16 | 50 | 1 | 500 | 4 | 5,000 | 2 |
| 32 | 3,000 | 1 | 7,000 | 1 | 8,000 | 46 |
| 64 | 5,000 | 1 | 7,000 | 3 | 8,000 | 131 |
| 128 | 7,000 | 1 | 8,000 | 3 | 8,000 | 248 |

With non-zero offsets, longer prefixes substantially improve memorization, suggesting that they could partially compensate for positive offsets. However, as illustrated by the red and green lines in Figure 3, even with extended prefix length, memorization at disadvantaged positions never fully matches the level observed at offset 0. Moreover, the 3B model achieves higher verbatim recall at offsets 64 and 128 compared to the 1B model, but memorization plateaus from 3B to 8B, indicating that larger models do not mitigate positional fragility and suffer similarly at suboptimal positions.

This inversion of optimal prefix length under offset conditions highlights a key property of verbatim memorization in realistic settings: it is strongly tied to positional cues, especially the beginning of the context window. Our findings align with the *"attention sink"* mechanism discovered by Xiao et al. [2024], where initial tokens disproportionately influence attention distribution and serve as computational anchors. Nasr et al. [2025] has also shown that the token that represents the document boundary, such as end-of-document (EOD) marker, can trigger memorization (in our case BOD). This raises a key question: is the verbatim memorization driven by the positional role of the initial tokens in general or by the special representational status of the BOD token itself? To disentangle these two factors, we design controlled ablation experiments that isolate the BOD token's contribution to verbatim recall.

## 4.2 BOD or Initial Tokens: An Ablation Study

We isolate the impact of the BOD token by retraining our 1B model with its attention masked, thereby effectively removing it from the input. In this setting, if memorization persists, it suggests that positional cues alone are sufficient; if it degrades, the BOD token likely serves as a dedicated retrieval anchor. To explore whether this change interacts with positional fragility, we evaluated memorization performance at three context-window offsets: 0, 50, and 100 tokens.

Figure 4 illustrates the impact of masking the BOD token on the model's memorization capability at the 1B scale. At offset 0 (left panel), where the prefix begins at the start of the context window, masking the BOD token yields negligible differences in verbatim recall. This suggests that the initial textual content alone can serve as an effective cue for memorization without an explicit BOD.

Interestingly, masking the BOD token leads to moderately higher verbatim recall in high-frequency buckets under offset conditions. At a 50-token offset, the BOD-masked model achieves up to 8% higher Rouge-L scores for prefixes between 750–2000 tokens—equivalent to roughly 40 additional tokens recalled verbatim. This suggests that, without an explicit BOD, the model leans more on local context, making it slightly more robust when recalling sequences from non-initial positions. Nonetheless, it continues to exhibit positional fragility (see Appendix D.1).

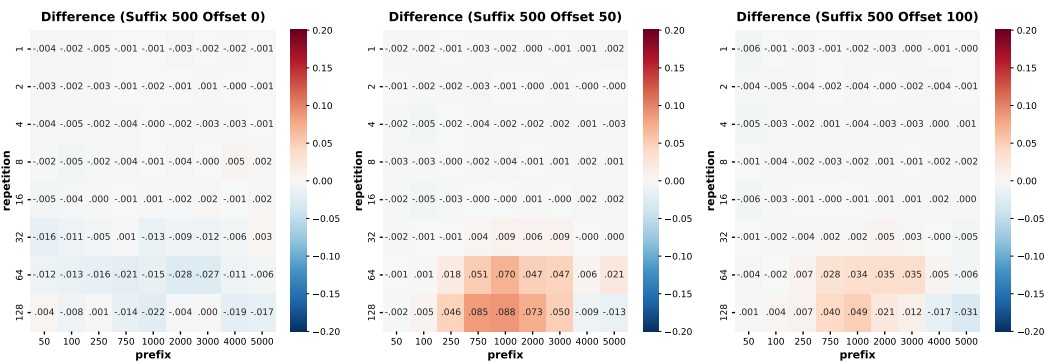

Figure 4: Effect of isolating the BOD token on the memorization capability of the 1B model measured by Rouge-L scores, computed over 500-token suffixes conditioned on varying prefix lengths (50–5000 tokens). Subplots from left to right correspond to context window offsets of 0, 50, and 100.

To ensure that this improvement in recall is not simply a byproduct of degraded generation quality—a symptom of degeneration discussed in §5—we evaluated the model using the MAUVE metric at longer prefix lengths (1000–5000 tokens), corresponding to regions with increased memorization. We also assessed downstream performance (see Table D.1 in the appendix). In both cases, the BOD-masked model performed on par with the baseline, indicating that increased recall does not come at the cost of linguistic ability.

## 5 Memorization: Under-Explored Cause of Text Degeneration

Our experiments reveal a connection between memorization capabilities and text degeneration in language models. While previous research observed repetitive patterns in training data as the primary cause of degeneration, highlighting a "repetition in, repetition out" phenomenon [Li et al., 2023], our findings suggest that limitations in how models encode and retrieve memorized information represent a more fundamental and previously unexplored mechanism underlying text degeneration. Rather than attributing repetitive outputs solely to repetitive training data, we demonstrate that deficiencies in memorization can also lead to text degeneration.

### 5.1 Context Window Offsets Reduces Output Text Quality

We find that the position of prefixes within the model's context window during pretraining influences the quality of the generated suffixes. As shown in Fig. 1, in the dense Gutenberg setting, where the model is trained on 10K sequences for 80 epochs, we observe that verbatim memorization effectively vanishes, while language modelling capabilities remain stable at larger offsets. In contrast, under the sparse Gutenberg setting with FM-probe bucketing, the model's output **degrades** in both lexical diversity and language coherence. An example is shown in Fig. 10.

This complements the "repetition in, repetition out" hypothesis by Li et al. [2023]. Their work specifically revealed a strong correlation between the repetition rate of 2-gram in training data and the repetitive generation in the model's output. Nevertheless, our results suggest that text degeneration may not be solely driven by repetition in training data.

### 5.2 Lower Exposure Frequency Reduces Output Text Quality

In addition to context window offsets, the training frequency of a sequence during training independently influences lexical diversity and language coherence. As illustrated in Fig. 5, sequences seen less frequently exhibit pronounced degradation in the model's linguistic capabilities across all model scales (Examples are shown in 11). However, larger models demonstrate greater resilience, achieving higher TTR and MAUVE scores than smaller counterparts, particularly at low frequencies, indicating that model scale modestly enhances linguistic capability.

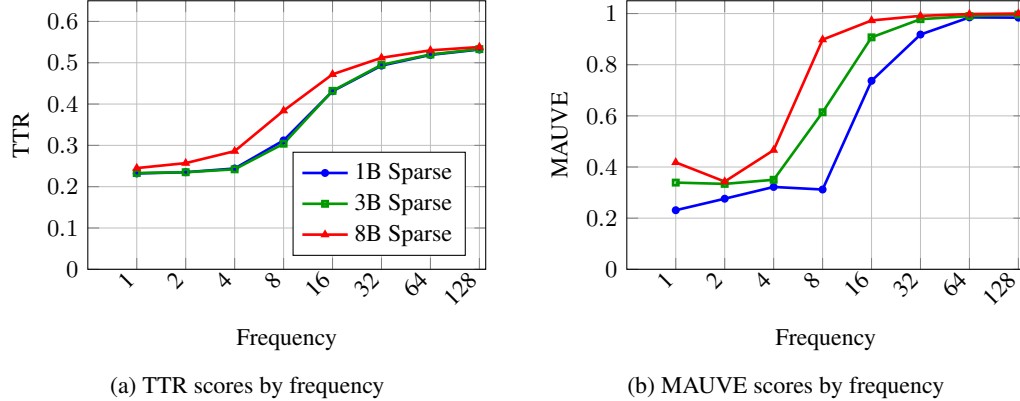

|                          |                          |
|:------------------------:|:------------------------:|
| (a) TTR scores by frequency | (b) MAUVE scores by frequency |

Figure 5: Impact of exposure frequency on text quality metrics for Llama models of different sizes, measured with two complementary metrics: (a) TTR scores quantify lexical diversity, showing improvement as training frequency increases; the ground truth TTR lies in the range 0.535–0.541;(b) MAUVE scores capture overall language quality and coherence, reaching near-perfect scores at high frequencies. All experiments used 500-token prefixes and were evaluated on 500-token suffixes with a zero offset.

## 5.3 Insufficient Memory Retrieval: A contributing Factor to Text Degeneration

Our empirical findings from §5.1 and §5.2 converge on a nuanced understanding of text degeneration in language models. Specifically, they reveal that imbalances in training data frequency can lead to distinct generation issues. High-frequency exposure often results in verbatim memorization, where the model reproduces training data verbatim. Conversely, low-frequency exposure does not necessarily enhance diversity; instead, it can exacerbate degeneration, manifesting as repetitions or thematic loops. This suggests a link between limited memory retrieval capabilities and degenerative text patterns.

The analysis of context window offsets further supports this relationship. As models attempt to continue text from distant positions, they tend to fall into thematic looping. This position-dependent degradation in linguistic ability supports that insufficient memory retrieval can cause text degeneration, particularly when memorisation targets must compete with diverse training data. This connection between retrieval limitations and text degeneration warrants further investigation in future work.

## 6  Probing the Robustness of Positional Fragility

Our primary analysis revealed positional fragility using greedy decoding on models trained from scratch. To ensure that this phenomenon reflects an inherent model behavior rather than an experimental artifact, we further examine its robustness across decoding strategies and training paradigms.

**Robustness Across Decoding Strategies**    To evaluate the robustness of positional fragility under alternative decoding configurations, we re-ran inference on the repetition-128 FM-Probe bucket using our 8B model under the *Sparse Gutenberg* setting with two additional decoding methods: (1) beam search ($k=5$), reporting the top-ranked hypothesis, and (2) nucleus sampling (top-$p=0.9$). Table 2 shows that beam search, a deterministic probability-maximizing strategy, achieves nearly perfect memorization at shallow offsets (e.g., ROUGE-L = 0.999 at offset 0). However, when recall fails at deeper offsets, it exhibits the most severe degeneration, supporting our claim that failed retrieval leads to incoherent outputs. In contrast, nucleus sampling yields lower memorization at shallow offsets but shows a much milder decline in lexical diversity at large offsets (e.g., TTR = 0.460 at offset 2048), suggesting that decoding strategy influences the severity of degeneration while leaving the offset-dependent pattern of positional fragility unchanged.

**Persistence Under Continued Pretraining**    We further examine whether positional fragility persists under a continued pretraining paradigm. Starting from the official *Llama 3.2 1B* model pretrained on 9T tokens, We continued pretraining using otherwise identical settings to the *Sparse Gutenberg* configuration, but with a smaller learning rate $1 \times 10^{-5}$. As shown in he continued-pretrained 1B

model exhibits slightly stronger memorization (higher ROUGE-L) and more diverse generations (higher TTR) than the from-scratch 1B model, while preserving the same offset-dependent decay pattern, confirming that positional fragility holds across training paradigms.

Table 2: **Robustness of positional fragility across decoding methods and training paradigms.** Top: Decoding strategies on the 8B model. Bottom: 1B continued pretrained model with greedy decoding. All evaluations conducted on the repetition-128 FM-Probe bucket using 500-token prefixes to generate 500-token suffixes. Across all settings, memorization (ROUGE-L) consistently declines with increasing offset, confirming that positional fragility persists regardless of decoding method or training paradigm.

| Decoding Strategies (8B Model) | ROUGE-L ↓ | | | TTR ↑ | | |
|---|---|---|---|---|---|---|
| Offset | Greedy | Beam | Nucleus | Greedy | Beam | Nucleus |
| 0 | 0.965 | 0.999 | 0.877 | 0.538 | 0.541 | 0.536 |
| 8 | 0.864 | 0.951 | 0.701 | 0.519 | 0.524 | 0.523 |
| 32 | 0.652 | 0.800 | 0.495 | 0.474 | 0.469 | 0.503 |
| 128 | 0.330 | 0.507 | 0.247 | 0.391 | 0.361 | 0.477 |
| 512 | 0.208 | 0.244 | 0.186 | 0.305 | 0.237 | 0.461 |
| 2048 | 0.192 | 0.186 | 0.180 | 0.279 | 0.196 | 0.460 |

| Training Paradigms (1B Model) | ROUGE-L ↓ | | TTR ↑ | | |
|---|---|---|---|---|---|
| Offset | Scratch | Continue | Scratch | Continue | |
| 0 | 0.744 | 0.669 | 0.521 | 0.521 | |
| 8 | 0.463 | 0.569 | 0.491 | 0.498 | |
| 32 | 0.346 | 0.426 | 0.465 | 0.482 | |
| 128 | 0.219 | 0.249 | 0.411 | 0.437 | |
| 512 | 0.191 | 0.202 | 0.345 | 0.364 | |
| 2048 | 0.185 | 0.188 | 0.293 | 0.340 | |

# 7 Leveraging Positional Fragility to Mitigate Verbatim Memorization

To demonstrate that positional fragility can be leveraged to mitigate verbatim memorization, we designed a Proof-of-concept (POC) experiment which we refer to as **Swapped Gutenberg**. For each sequence in our FM-Probe buckets, we replace the first 4,000 tokens (the *swapped part*) with tokens from randomly selected Project Gutenberg excerpts within our 10,000-sequence corpus. Thus, the initial segment becomes diverse and contextually unrelated, while the remainder (the *retained part*) maintains controlled repetition frequencies (1 to 256 repetitions). Each sequence includes only one initial BOS token, with no additional special tokens between swapped and retained segments (see Fig. 2). We train 1B and 8B models and evaluate verbatim memorization using prefixes from the retained part.

Table 3 shows that the principle demonstrated in this POC works surprisingly well regardless of model scale, keeping verbatim recall consistently low even at the highest exposure frequency. ROUGE-L and LCCS scores remain near the baseline similarity between unrelated texts across all frequencies, indicating **little to no extractable memorization**. As repetition increases, perplexity steadily declines, particularly in the 8B model, yet without corresponding gains in verbatim overlap. Both TTR and MAUVE remain stable, with the 8B model producing more coherent and lexically rich outputs. These results confirm that shifting sensitive content away from the context window's beginning is a simple but effective principle for mitigation at all scales.

Our POC mitigation strategy also neutralizes the offset effects entirely. Recall that this effect manifests in two ways: (1) shorter prefixes are most effective at triggering memorization when taken from the beginning of the context window, and (2) even short prefixes lose their efficacy when shifted to later positions, though longer prefixes can partially compensate. However, both effects disappear when the memorization target is displaced 4,000 tokens deep, as in the Swapped Gutenberg setup. Table 9 shows that increasing prefix length no longer improves memorization, and Table 10 reveals consistent recall across offsets from 0 to 2048 tokens.

Table 3: Comparison of text generation metrics under Swapped and Sparse Gutenberg settings. Each row reports performance on 500-token suffixes generated from 500-token prefixes at offset 0. Selected exposure frequencies (1, 8, 64, 128) are shown to highlight transitions in memorization behavior. Rows 1–4 correspond to Swapped Gutenberg, where prefixes are offset relative to the *retained part* of the sequence; rows 5–8 correspond to standard Sparse Gutenberg. Full results for each setting and frequency range are provided in Table 11 in the Appendix.

| Freq. | Rouge-L↓ | | LCCS↓ | | Perplexity | | TTR↑ | | MAUVE↑ | |
|---|---|---|---|---|---|---|---|---|---|---|
| | 1B | 8B | 1B | 8B | 1B | 8B | 1B | 8B | 1B | 8B |
| 1 | 0.178 | 0.176 | 0.009 | 0.009 | 40.793 | 82.206 | 0.348 | 0.467 | 0.779 | 0.970 |
| 8 | 0.180 | 0.179 | 0.008 | 0.009 | 30.621 | 12.841 | 0.350 | 0.467 | 0.805 | 0.934 |
| 64 | 0.182 | 0.184 | 0.009 | 0.011 | 16.356 | 3.670 | 0.377 | 0.469 | 0.851 | 0.905 |
| 128 | 0.181 | 0.181 | 0.009 | 0.011 | 16.017 | 3.636 | 0.377 | 0.468 | 0.911 | 0.958 |
| 1 | 0.181 | 0.185 | 0.008 | 0.009 | 26.036 | 16.089 | 0.225 | 0.245 | 0.231 | 0.418 |
| 8 | 0.183 | 0.191 | 0.008 | 0.016 | 12.698 | 3.430 | 0.231 | 0.384 | 0.312 | 0.898 |
| 64 | 0.522 | 0.888 | 0.415 | 0.858 | 1.125 | 1.023 | 0.497 | 0.530 | 0.985 | 0.998 |
| 128 | 0.744 | 0.965 | 0.682 | 0.951 | 1.051 | 1.012 | 0.521 | 0.538 | 0.984 | 1.000 |

This disappearance results from two key design choices: the swapped segment is randomly sampled across sequences that break fixed positional associations; and the memorization target is embedded far from the model's typical retrieval anchor near the beginning of the context window.

Furthermore, this method not only suppresses memorization but we also find that it improves generation quality. Unlike the sparse Gutenberg setting, where prompting from offset positions leads to lexical degradation and thematic looping, Swapped Gutenberg maintains high coherence and diversity across all conditions. As shown in Table 3, MAUVE and TTR scores remain stable and high, in stark contrast to the quality collapse observed in Sparse Gutenberg setting (also see Figure 1b and 5). These findings highlight a central implication of our study: by decoupling sensitive content from early-context anchors, it is possible to suppress extractable memorization while preserving generation quality and downstream task performance (see Table 12) across model scales and training regimes. This provides a compelling justification for future research into developing more practical, production-ready mitigation techniques based on this principle. Examples are shown in Fig. 12.

## 8   Batch Size Impact

Our experiments reveal a clear advantage to pretraining language models with smaller batch sizes, which necessitates more iterations given a fixed compute budget. This advantage manifests as reduced verbatim recall (Fig. 6a), improved language coherence (Fig. 6b), and enhanced performance across most downstream benchmarks (Appendix D.2). Notably, these improvements emerge despite maintaining both a constant proportion of Gutenberg sequences in each batch and consistent total training tokens across all experiments. To the best of our knowledge, this has not been shown in prior work.

## 9   Related Work

**Factors Known to Affect Memorization**   Several factors are known to influence a model's tendency to memorize and reproduce training data verbatim. (1) Model scale: larger models tend to memorize more due to their greater capacity to fit specific examples [Carlini et al., 2021, Lesci et al., 2024]. (2) Repetition frequency: memorization probability increases roughly log-linearly with the number of times a sequence appears during training [Carlini et al., 2023]; we confirm both effects in our setting (Appendix C.1). (3) Complexity: complex sequences seen only once can be latently memorized and retrieved via small parameter perturbations [Duan et al., 2025]. (4) Timing of exposure: sequences seen later in training are more likely to be retained than those encountered earlier [Kiyomaru et al., 2024, Lesci et al., 2024]. (5) Prefix length: while longer prefixes improve recall in prior work [Carlini et al., 2023], we find this effect depends heavily on the prefix's position within the context window (§ 4.1).

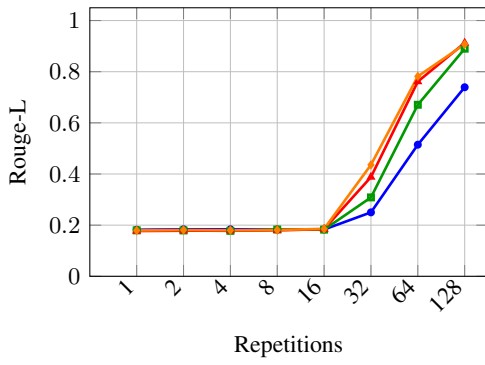
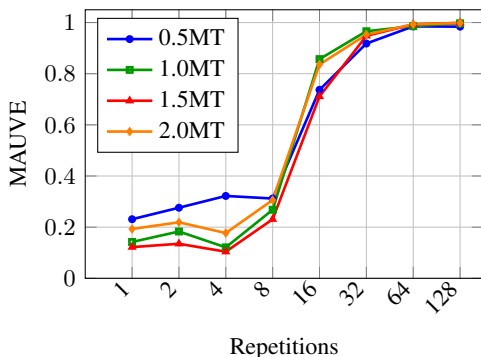

(a) Rouge-L scores by repetition frequency

(b) MAUVE scores by repetition frequency

Figure 6: Impact of batch size on verbatim memorization and language coherence for 1B models, as measured by Rouge-L (left) and MAUVE scores (right) across repetition frequencies. Models were trained with batch sizes ranging from 0.5MT to 2.0MT, maintaining constant total training tokens. As batch size increases, we observe enhanced language coherence (MAUVE) at low repetition frequencies, while verbatim memorization (Rouge-L) increases most significantly for sequences seen more than 32 times. The gap between smallest and largest batch sizes reaches up to 0.15 in Rouge-L score at high repetition frequencies, corresponding to approximately 90 fewer tokens being memorized verbatim in the 500-token continuations. This suggests smaller batch sizes with more update steps reduce exact memorization while preserving language quality.

**Preventing Verbatim Memorization During Pretraining** Pretraining-based strategies have demonstrated promising results in reducing verbatim memorization without sacrificing model performance. One prominent strategy is data deduplication, which removes duplicate or near-duplicate sequences from the training corpus. Lee et al. [2022] showed that deduplication substantially reduces memorization while preserving downstream performance. Kandpal et al. [2022] further demonstrated that it lowers the success rate of extraction attacks. Another promising strategy is the recently proposed Goldfish Loss [Hans et al., 2025], which selectively masks tokens within n-gram windows during training. By consistently omitting these tokens from the loss computation, it prevents the model from learning exact token-to-context mappings, thereby disrupting extractable memorization while retaining language modeling capabilities.

## 10 Conclusion

We demonstrate that LLMs exhibit a pronounced positional bias in their memorization behavior, with verbatim recall most easily triggered by prefixes near the start of the context window. This offset effect not only distorts standard memorization risk assessments but also offers a new mitigation pathway: simply shifting sensitive content deeper into the context suppresses extractable memorization without degrading generation quality. Our findings further reveal that when retrieval fails—due to offset or insufficient exposure—models often degenerate into repetitive, low-diversity output, linking memorization to generation stability. These insights position offset as a critical and previously underexplored axis in the study of memorization, and suggest simple, scalable mitigation strategies that complement existing techniques. Limitations of our study and directions for future work are discussed in Appendix B.

## Acknowledgements

This work was conducted during Yixuan Xu's Master thesis at the ETH AI Center, while he was a Master student at EPFL. Yixuan would like to thank HPC expert Antoni-Joan Solergibert i Llaquet for his technical guidance. This work was supported as part of the Swiss AI Initiative by a grant from the Swiss National Supercomputing Centre (CSCS) under project ID a06 on Alps.

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

# A   Implementation Details

## A.1   Computing Infrastructure

All experiments were conducted on the Alps supercomputer at the Swiss National Supercomputing Centre (CSCS).[2] Each compute node includes four NVIDIA GH200 Grace Hopper Superchips, with each GH200 integrating a 72-core ARM-based Grace CPU and a Hopper H100 GPU, interconnected via NVLink-C2C. Each GPU provides 96 GB of CUDA memory.

**Pretraining**   For most of our pretrain runs, we utilized a 15-node configuration. Each GPU processed a maximum micro-batch size of 3 sequences, each consisting of 8,192 tokens. Only one specific experiment, involving the 1B model trained with a global batch size of 240 sequences, required a larger setup of 30 nodes. Pretraining of the 1B, 3B, and 8B LLaMA models was orchestrated exclusively using Data Parallelism (DP). Given that the 8B model parameters comfortably fit into the memory of a single GH200 GPU, neither tensor parallelism nor pipeline parallelism was necessary for efficient training. The throughput and compute for each model scale are summarized in Table 4.

Table 4: Aggregate pretraining throughput (tokens per second across all 60 GPUs) and total GPU hours required for pretraining each model scale.

| Model Scale | Throughput (tokens/s) | GPU Hours |
|---|---|---|
| 1B | 2,308,935 | ∼600 |
| 3B | 936,003 | ∼1,440 |
| 8B | 450,017 | ∼3,060 |

**Inference & Evaluation**   Inference Time: On a single node, processing one FM-probe bucket (500-token prefix and 500-token suffix) takes ∼1.5 minutes for the 1B model and ∼3 minutes for the 8B model, including both text generation and evaluation (Perplexity, LCCS, EM, ROUGE-L, TTR). MAUVE is computed separately and takes ∼34 minutes per bucket of the same size on a single GH200.

## A.2   Model

We pretrain our models using an adapted version of NVIDIA Megatron-LM [Shoeybi et al., 2020], incorporating modifications described in the main paper. Table 5 details the architectural specifications and training hyperparameters of our LLaMA-based models. The tokenizer we utilised is the `OpenMath2-Llama3.1-8B`[3].

## A.3   Reproducibility

To enhance reproducibility, we fix the global random seed to 42 across all experiments. However, this alone does not ensure strict determinism in deep learning workflows. In particular, using different batch sizes may trigger different optimized CUDA kernels, leading to minor numerical variations and non-deterministic outputs despite identical seeds and initialization [4]. For guaranteed reproducibility, running inference and downstream evaluation with a batch size of 1 on the same GPU type (NVIDIA GH200) using our provided scripts yields identical results. Due to time constraints and the scale of our experiments, we instead use a batch size of 20 for model generation to maximize throughput, and 4 for downstream evaluation.

---

[2] `https://www.cscs.ch/computers/alps/`
[3] `https://huggingface.co/nvidia/OpenMath2-Llama3.1-8B`
[4] `https://pytorch.org/docs/stable/notes/numerical_accuracy.html`

Table 5: Architectural specifications and training hyperparameters of the LLaMA-based models. Hyper-parameters are the default value from Megatron-LM.

| Parameter | 1B | 3B | 8B |
|---|---|---|---|
| Shared Input-Output Projections | Yes | Yes | No |
| Hidden Size | 2,048 | 3,072 | 4,096 |
| Intermediate Size | 8,192 | 8,192 | 14,336 |
| Number of Layers | 16 | 28 | 32 |
| Number of Attention Heads | 32 | 24 | 32 |
| Number of Key-Value Heads | 8 | 8 | 8 |
| Head Dimension | 64 | 128 | 128 |
| RoPE scaling factor | 32.0 | 32.0 | 8.0 |
| RoPE base frequency | | 500,000 | |
| Max Positional Embeddings | | 131,072 | |
| Vocabulary Size | | 128,256 | |
| Warmup Steps | 2,000 | 2,000 | 200 |
| Initial Learning Rate | $3 \times 10^{-4}$ | $3 \times 10^{-4}$ | $2.2 \times 10^{-4}$ |
| Weight Decay | 0.01 | 0.01 | 0.1 |
| Min. Learning Rate | $3 \times 10^{-5}$ | $3 \times 10^{-5}$ | $2.2 \times 10^{-5}$ |
| Batch Size | | 60 | |
| Optimizer | | Adam | |

# B    Limitations and Future Directions

## B.1    Limitations

While our proof-of-concept mitigation validates the principle that positional displacement can suppress extractable memorization, it assumes prior knowledge of which training segments are sensitive and thus does not establish a directly actionable workflow for large-scale pretraining pipelines. Future work may explore how to operationalize this principle—e.g., by combining positional fragility with automated sensitivity detection or data-curation strategies—to mitigate memorization at scale in realistic training environments.

Moreover, our corpus, while comprising approximately 83B tokens, remains several orders of magnitude smaller than real-world foundation model datasets trained on trillions of tokens. As such, our findings may not fully capture the memorization dynamics present at production scale.

Finally, our study lacks formal privacy or generalization guarantees (e.g., differential privacy [Anil et al., 2022]) and is limited to left-to-right autoregressive models trained from scratch. The empirical scope does not extend to fill-in-the-middle or retrieval-augmented architectures, which may exhibit distinct positional behaviors.

## B.2    Future Directions

Our analysis is limited to conventional *left-to-right autoregressive models* following the LLaMA architecture. We do not study models trained with *Fill-in-the-Middle (FIM)* objectives, such as OpenAI Codex [Bavarian et al., 2022] and DeepSeek-V3 [DeepSeek-AI et al., 2025], which reconstruct missing spans from both preceding and following context. This bidirectional formulation enables flexible anchoring and contextual recombination, and is conceptually similar to our mitigation strategy of displacing sensitive content deeper in the context window. We leave the study of *positional fragility* and *memorization* in FIM-trained models to future work.

More broadly, it remains an open question whether similar offset-sensitive memorization behaviors emerge in non-text modalities such as code, images, or audio, and whether comparable mitigation

strategies are effective. Extending offset-aware analysis to these domains may shed light on the generality of positional fragility and its broader implications for memorization and generation quality.

## C  Reproducing and Extending Prior Work

### C.1  Exposure Frequency & Model Size: Interconnected Memorization Dynamics

As shown in Figure 7a, our analysis reveals a significant inverse relationship between model scale and memorization threshold. Following the vertical axis (repetition frequency), the 1B model requires approximately 64 exposures to demonstrate substantial verbatim recall of suffix tokens. This threshold decreases to approximately 32 exposures for the 3B model, and further reduces to only 16 exposures for the 8B model. This systematic pattern suggests that memorization efficiency scales inversely with model size. Extrapolating this trend, a 70B model, without specific memorization mitigation strategies, might memorize sequences after just 2-3 exposures.

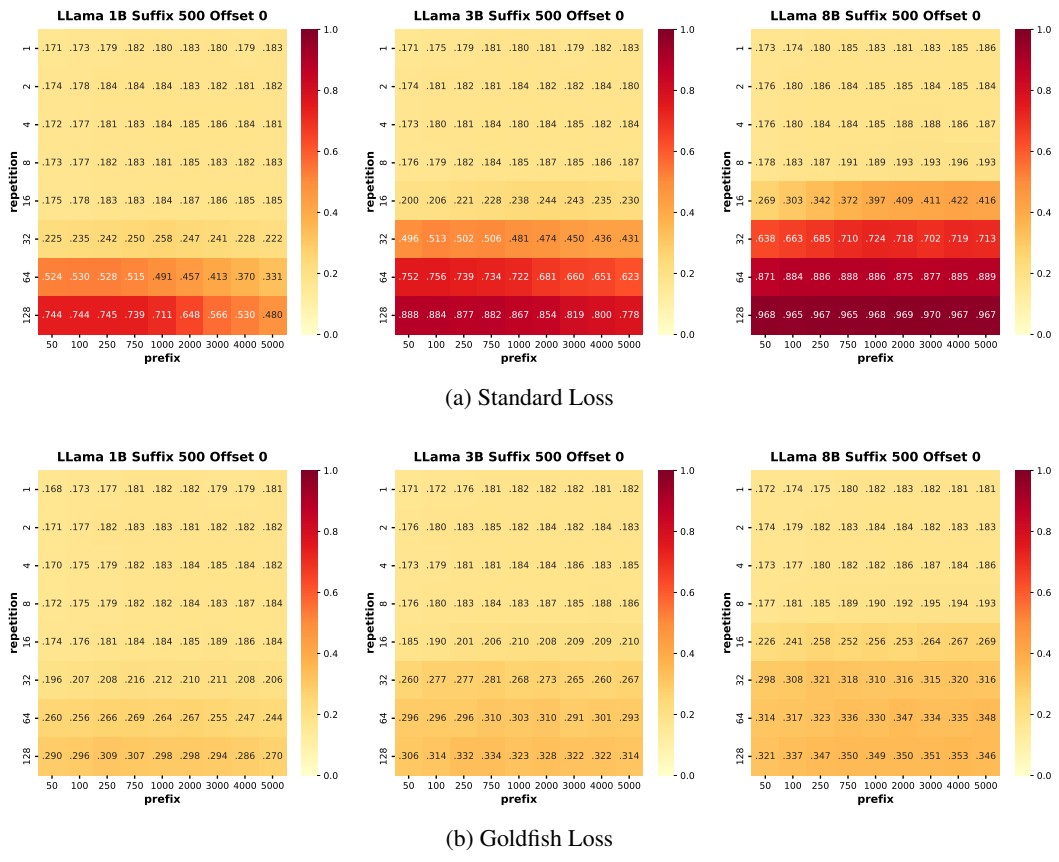

Figure 7: Comparison of verbatim memorization across language models (1B, 3B, and 8B) with varying prefix lengths under Sparse Gutenberg Scenario. The heatmaps display Rouge-L scores computed for 500-token suffixes at offset 0, across different prefix lengths (50–5000 tokens, x-axis) and repetition frequencies (1–128, y-axis).. By comparing Figures 7a and 7b, we observe that Goldfish Loss effectively reduces verbatim recall. Nevertheless, despite employing Goldfish Loss, larger models or higher exposure frequencies still exhibit trends toward memorizing training data.

Complementing this frequency effect, the horizontal axis (prefix length) reveals how different-sized models leverage contextual information. For smaller models (1B and 3B), shorter prefixes of around 50 tokens triggered the strongest verbatim memorization, supporting the hypothesis that initial tokens disproportionately influence memorization patterns. In contrast, the 8B model maintained consistent memorization even with longer prefixes, demonstrating that larger models develop more sophisticated contextual memory mechanisms that operate effectively across varied input lengths.

These dual observations regarding decreased repetition requirements and enhanced contextual processing in larger models together demonstrate that scaling fundamentally transforms memorization capabilities. This explains why larger language models demonstrate both improved generalization and increased memorization risk: their enhanced capacity enables them to memorize more efficiently while simultaneously developing more nuanced representations of linguistic patterns. Our results complement to [Carlini et al., 2023].

## C.2 Goldfish Loss

We reproduce the extreme and standard memorization scenarios originally proposed by Hans et al. [2025], adapting them to our dense and sparse Gutenberg experiments, respectively. However, our implementation substantially scales up the original design by pretraining from scratch with sequences that are 4× longer and drawn from 100× more documents. We use the dense Gutenberg setting to tune the optimal Goldfish hyperparameters for the sparse Gutenberg, yielding a dropout ratio of 2% ($k = 50$) and a context width ($h$) of 50 tokens.

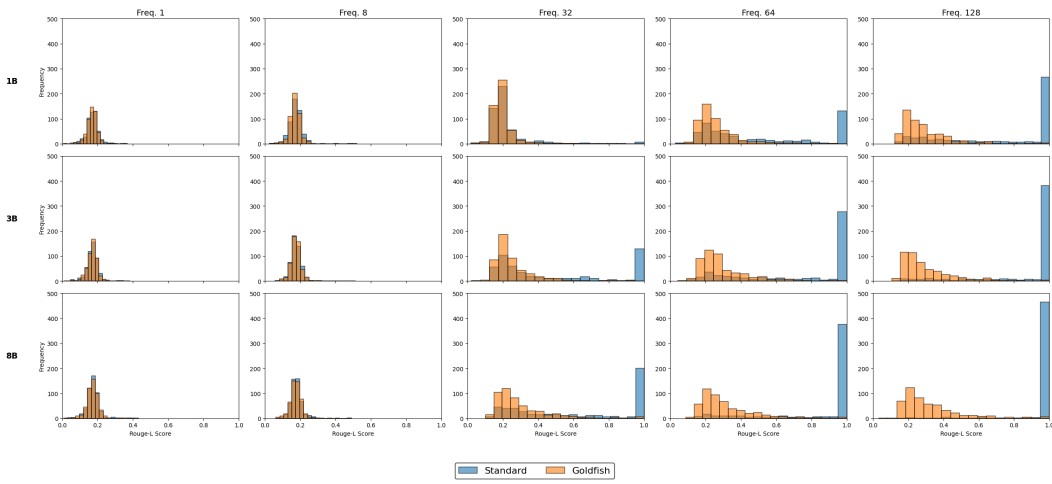

Figure 8: Comparison of Rouge-L distributions between standard and Goldfish Loss LLaMA models of sizes 1B, 3B, and 8B, evaluated with a 50-token prefix and a 500-token suffix at offset 0. The histograms display Rouge-L overlap scores across varying training exposures (1, 8, 32, 64, and 128).

The sparse Gutenberg results confirm that Goldfish Loss effectively mitigates verbatim memorization across all model sizes. As shown in Figure 7b and Figure 8, models trained without memorization mitigation and evaluated with zero positional offset exhibit pronounced verbatim recall abilities that systematically increase with both exposure frequency and model scale. The standard-trained 8B model achieves Rouge-L scores above 0.95 for most test sequences after 128 exposures, while Goldfish Loss models maintain scores consistently below 0.4 even under the most favorable memorization conditions.

Table 6: Downstream Task Performances: Despite training with 2% tokens being dropped, model trained with goldfish loss still shows comparable even superior downstream performance then model trained with standard cross-entropy loss.

| Model | Wiki. | Hella. | | ARC-c | | ARC-e | | PIQA | Wino. | CSQA | MMLU |
|---|---|---|---|---|---|---|---|---|---|---|---|
| | ppl↓ | acc↑ | norm↑ | acc↑ | norm↑ | acc↑ | norm↑ | acc↑ | acc↑ | acc↑ | acc↑ |
| Standard 1B | 18.71 | 40.43 | 52.31 | **33.36** | **35.15** | **68.10** | 63.13 | 71.00 | **53.91** | **21.79** | 23.65 |
| Goldfish 1B | 18.96 | **40.44** | **52.41** | 32.08 | 32.25 | 67.68 | **63.38** | **71.11** | 53.43 | 19.00 | **25.10** |
| Standard 3B | 15.42 | **46.13** | **59.93** | **38.40** | **40.44** | **73.65** | **68.01** | **73.99** | 57.06 | **21.87** | **25.69** |
| Goldfish 3B | **15.01** | 46.01 | 59.89 | 36.52 | 40.10 | 71.84 | 67.76 | 73.72 | **58.41** | 20.72 | 25.42 |
| Standard 8B | 13.15 | 49.74 | 65.74 | 42.24 | 45.99 | 75.97 | 72.18 | 75.52 | 61.88 | **20.56** | 24.53 |
| Goldfish 8B | **12.44** | **50.29** | **66.61** | **43.00** | **46.67** | **76.89** | **73.78** | **75.63** | **62.43** | 20.39 | **26.98** |

Beyond verbatim memorization examination, the sparse Gutenberg framework reveals that Goldfish Loss effectively prevents verbatim memorization without compromising downstream performance. In fact, models trained with Goldfish Loss maintain competitive performance across all benchmarks, with this advantage becoming more pronounced as model scale increases. The 8B Goldfish model demonstrates this trend most clearly, outperforming its standard counterpart on several tasks , as shown in Table 6. This exciting result demonstrating that pretraining-based memorization mitigation can simultaneously address copyright concerns and enhance model capabilities.

# D    Additional Results

## D.1    BOD or Initial Tokens Ablation

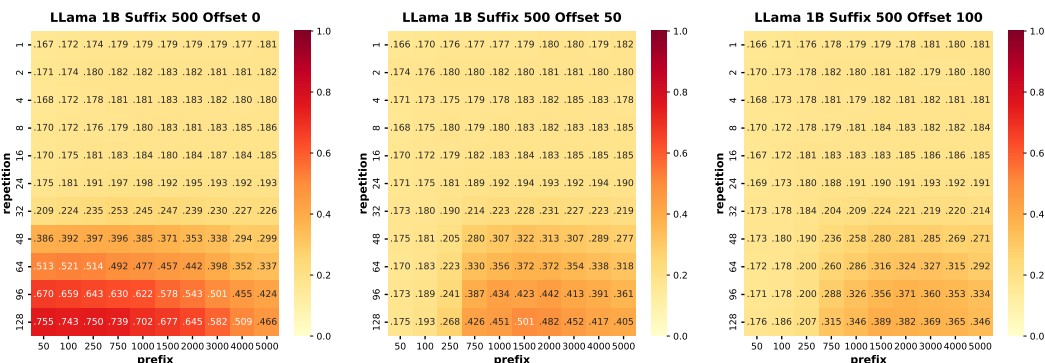

Figure 9: Verbatim memorization of the 1B model trained with `BOD` attention masked, under the Sparse Gutenberg setup. Heatmaps show Rouge-L scores for 500-token suffixes across prefix lengths (50–5000 tokens, x-axis) and repetition frequencies (1–128, y-axis), evaluated at offsets 0, 50, and 100 (left to right). Memorization degrades consistently as the offset increases, highlighting persistent positional fragility despite masking the `BOD` token.

Table 7: Downstream task performance of the baseline 1B model compared to its variant trained with `BOD` attention masked (`-BOD`).

| Model | Wiki. | Hella. | | ARC-c | | ARC-e | | PIQA | Wino. | CSQA | MMLU |
|---|---|---|---|---|---|---|---|---|---|---|---|
| | ppl↓ | acc↑ | norm↑ | acc↑ | norm↑ | acc↑ | norm↑ | acc↑ | acc↑ | acc↑ | acc↑ |
| Baseline | 18.71 | 40.43 | **52.31** | **33.36** | **35.15** | **68.10** | 63.13 | 71.00 | 53.91 | **21.79** | 23.65 |
| - BOD | **17.89** | **40.98** | 52.10 | 31.40 | 33.96 | 68.06 | **64.65** | **71.16** | **54.54** | 20.88 | **24.06** |

## D.2    Batch Size Impact on Downstream Performance

Larger batch sizes correspond to higher perplexity scores on validation data, with the 240 GBS (global batch size) model showing a perplexity of 22.86 compared to 18.71 for the 60 GBS model — a 22.2% degradation in predictive performance despite seeing identical training content. These same models demonstrate poorer performance as the batch size increases. The 60 GBS model consistently outperforms larger batch variants on reasoning-intensive benchmarks; for example, it achieves 33.36% accuracy on ARC-c compared to just 29.52% for the 240 GBS model, where the baseline random guess is 25%.

Our findings resonate with established patterns documented in deep learning research regarding batch size impact. When models train with smaller batches, the resulting gradient calculations introduce natural variability between updates. This inherent noise effectively serves as an implicit regularization mechanism, as each weight adjustment follows a slightly different trajectory, preventing overspecialization to training examples. This aligns with [Masters and Luschi, 2018] that reduced batch sizes promote more robust generalization through stochastic optimization pathways, ultimately yielding superior performance on evaluation tasks.

Table 8: Downstream task performance of models trained with varying batch sizes. Gradient updates indicate the number of optimization steps taken during pretraining.

| Model | Grad. Updates | Wiki. ppl↓ | Hella. acc↑ | Hella. norm↑ | ARC-c acc↑ | ARC-c norm↑ | ARC-e acc↑ | ARC-e norm↑ | PIQA acc↑ | Wino. acc↑ | CSQA acc↑ | MMLU acc↑ |
|---|---|---|---|---|---|---|---|---|---|---|---|---|
| 60 GBS | 170,005 | **18.71** | **40.43** | **52.31** | **33.36** | **35.15** | **68.10** | **63.13** | **71.00** | 53.91 | **21.79** | 23.65 |
| 120 GBS | 85,002 | 18.75 | 39.85 | 51.02 | 30.97 | 33.36 | 67.80 | 61.62 | 70.13 | **55.88** | 19.74 | 23.18 |
| 180 GBS | 56,668 | 20.74 | 39.62 | 49.83 | 32.25 | 34.22 | 67.76 | 62.54 | 69.86 | 53.20 | 19.98 | **24.78** |
| 240 GBS | 42,501 | 22.86 | 39.09 | 49.40 | 29.52 | 31.66 | 67.13 | 60.94 | 70.02 | 52.80 | 19.49 | 23.73 |

Viewed from another angle, our work shows that fewer parameter updates yield reduced model generalization and increased verbatim memorization. This finding aligns with [Hoffer et al., 2017], which attributes large batch performance deficits primarily to reduced update frequency rather than batch size itself. The considerably fewer parameter updates with larger batches (42,501 at 240 GBS compared to 170,005 at 60 GBS) may accelerate verbatim memorization while simultaneously compromising broader generalization capabilities.

## D.3   Swapped Gutenberg

Table 9: Impact of prefix length on text generation metrics for 1B and 8B models, evaluated on 500-token suffixes after 256 exposures at offset 0 under Sparse Gutenberg setting. The perplexity remains relatively stable across different prefix lengths, indicating that increasing prefix length does not substantially compensate for our method's effectiveness. For reference suffixes, the TTR ranges between 0.535 and 0.541.

| Prefix | Rouge-L↓ 1B | Rouge-L↓ 8B | LCCS↓ 1B | LCCS↓ 8B | Perplexity↓ 1B | Perplexity↓ 8B | TTR↑ 1B | TTR↑ 8B | MAUVE↑ 1B | MAUVE↑ 8B |
|---|---|---|---|---|---|---|---|---|---|---|
| 50 | 0.172 | 0.174 | 0.007 | 0.009 | 15.085 | 3.620 | 0.356 | 0.451 | 0.809 | 0.914 |
| 500 | 0.181 | 0.183 | 0.009 | 0.011 | 14.880 | 3.530 | 0.380 | 0.470 | 0.907 | 0.948 |
| 1000 | 0.181 | 0.183 | 0.009 | 0.011 | 15.682 | 3.464 | 0.386 | 0.472 | 0.914 | 0.985 |
| 2000 | 0.181 | 0.183 | 0.009 | 0.012 | 15.613 | 3.678 | 0.392 | 0.480 | 0.882 | 0.947 |
| 3000 | 0.184 | 0.188 | 0.010 | 0.014 | 15.625 | 3.844 | 0.398 | 0.476 | 0.934 | 0.945 |

Table 10: Impact of offset position on text generation metrics for 1B and 8B models after 256 exposures, evaluated on 500-token suffixes generated from corresponding 500-token prefixes. For reference suffixes, the TTR ranges between 0.537 and 0.541. Rouge-L and LCCS scores remain flat across all offsets and hover around the unrelated-text baseline, confirming that our mitigation neutralizes the offset effect. Generation quality (TTR, MAUVE) remains stable, indicating no degradation in fluency or coherence.

| Offset | Rouge-L↓ | | LCCS↓ | | Perplexity↓ | | TTR↑ | | MAUVE↑ | |
|---|---|---|---|---|---|---|---|---|---|---|
| | 1B | 8B | 1B | 8B | 1B | 8B | 1B | 8B | 1B | 8B |
| 0 | 0.181 | 0.183 | 0.009 | 0.011 | 14.880 | 3.530 | 0.380 | 0.470 | 0.907 | 0.970 |
| 1 | 0.182 | 0.182 | 0.009 | 0.011 | 14.862 | 3.525 | 0.384 | 0.471 | 0.862 | 0.959 |
| 2 | 0.182 | 0.183 | 0.009 | 0.011 | 14.866 | 3.518 | 0.384 | 0.469 | 0.939 | 0.947 |
| 4 | 0.181 | 0.183 | 0.009 | 0.011 | 14.847 | 3.526 | 0.379 | 0.466 | 0.833 | 0.959 |
| 8 | 0.181 | 0.182 | 0.009 | 0.011 | 14.846 | 3.512 | 0.381 | 0.470 | 0.900 | 0.968 |
| 16 | 0.182 | 0.184 | 0.009 | 0.012 | 14.773 | 3.534 | 0.379 | 0.469 | 0.876 | 0.963 |
| 32 | 0.182 | 0.183 | 0.009 | 0.011 | 14.775 | 3.573 | 0.373 | 0.468 | 0.876 | 0.951 |
| 64 | 0.180 | 0.183 | 0.009 | 0.011 | 14.843 | 3.528 | 0.373 | 0.464 | 0.899 | 0.971 |
| 128 | 0.181 | 0.183 | 0.009 | 0.011 | 15.047 | 3.621 | 0.378 | 0.468 | 0.900 | 0.975 |
| 256 | 0.181 | 0.183 | 0.009 | 0.011 | 15.385 | 3.763 | 0.375 | 0.466 | 0.870 | 0.948 |
| 512 | 0.181 | 0.182 | 0.009 | 0.011 | 15.806 | 3.917 | 0.368 | 0.466 | 0.934 | 0.937 |
| 1024 | 0.180 | 0.182 | 0.008 | 0.010 | 16.247 | 4.278 | 0.372 | 0.467 | 0.874 | 0.944 |
| 2048 | 0.181 | 0.183 | 0.009 | 0.011 | 16.619 | 4.595 | 0.373 | 0.466 | 0.934 | 0.962 |

Table 11: Impact of exposure frequency on text generation metrics for 1B and 8B models evaluated on 500-token suffixes generated from corresponding 500-token prefixes at offset 0. The upper section shows results under Swapped Gutenberg setting, while the lower section shows results under Sparse Gutenberg setting. For reference suffixes, the TTR ranges between 0.535 and 0.541.

| Freq. | Rouge-L↓ | | LCCS↓ | | Perplexity↓ | | TTR↑ | | MAUVE↑ | |
|---|---|---|---|---|---|---|---|---|---|---|
| | 1B | 8B | 1B | 8B | 1B | 8B | 1B | 8B | 1B | 8B |
| 1 | 0.178 | 0.176 | 0.009 | 0.009 | 40.793 | 82.206 | 0.348 | 0.467 | 0.779 | 0.970 |
| 2 | 0.179 | 0.176 | 0.008 | 0.008 | 40.710 | 63.966 | 0.349 | 0.467 | 0.854 | 0.969 |
| 4 | 0.178 | 0.178 | 0.008 | 0.009 | 35.502 | 34.562 | 0.350 | 0.462 | 0.784 | 0.910 |
| 8 | 0.180 | 0.179 | 0.008 | 0.009 | 30.621 | 12.841 | 0.350 | 0.467 | 0.805 | 0.934 |
| 16 | 0.181 | 0.182 | 0.009 | 0.010 | 24.063 | 5.008 | 0.355 | 0.466 | 0.759 | 0.965 |
| 32 | 0.181 | 0.181 | 0.009 | 0.011 | 18.585 | 3.970 | 0.374 | 0.470 | 0.940 | 0.958 |
| 64 | 0.182 | 0.184 | 0.009 | 0.011 | 16.356 | 3.670 | 0.377 | 0.469 | 0.851 | 0.905 |
| 128 | 0.181 | 0.181 | 0.009 | 0.011 | 16.017 | 3.636 | 0.377 | 0.468 | 0.911 | 0.958 |
| 256 | 0.181 | 0.183 | 0.009 | 0.011 | 14.880 | 3.530 | 0.380 | 0.470 | 0.907 | 0.948 |
| 1 | 0.181 | 0.185 | 0.008 | 0.009 | 26.036 | 16.089 | 0.225 | 0.245 | 0.231 | 0.418 |
| 2 | 0.182 | 0.184 | 0.008 | 0.009 | 24.474 | 13.902 | 0.227 | 0.257 | 0.276 | 0.343 |
| 4 | 0.184 | 0.184 | 0.008 | 0.009 | 19.065 | 9.047 | 0.228 | 0.286 | 0.322 | 0.466 |
| 8 | 0.183 | 0.191 | 0.008 | 0.016 | 12.698 | 3.430 | 0.231 | 0.384 | 0.312 | 0.898 |
| 16 | 0.183 | 0.372 | 0.010 | 0.232 | 5.741 | 1.229 | 0.323 | 0.472 | 0.737 | 0.973 |
| 32 | 0.250 | 0.710 | 0.086 | 0.637 | 1.565 | 1.054 | 0.443 | 0.512 | 0.918 | 0.991 |
| 64 | 0.522 | 0.888 | 0.415 | 0.858 | 1.125 | 1.023 | 0.497 | 0.530 | 0.985 | 0.998 |
| 128 | 0.744 | 0.965 | 0.682 | 0.951 | 1.051 | 1.012 | 0.521 | 0.538 | 0.984 | 1.000 |

Table 12: Downstream Task Performances for Swapped Gutenberg experiments.

| Model | Wiki. | Hella. | | ARC-c | | ARC-e | | PIQA | Wino. | CSQA | MMLU |
|---|---|---|---|---|---|---|---|---|---|---|---|
| | ppl↓ | acc↑ | norm↑ | acc↑ | norm↑ | acc↑ | norm↑ | acc↑ | acc↑ | acc↑ | acc↑ |
| Swapped 1B | 22.31 | 41.13 | 52.22 | 31.83 | 34.56 | 68.73 | 63.51 | 71.93 | 55.01 | 20.56 | 23.27 |
| Swapped 8B | 15.71 | 49.41 | 65.48 | 42.06 | 44.11 | 75.88 | 71.93 | 75.79 | 61.09 | 20.64 | 25.89 |

# E  License for Existing Assets

**Project Gutenberg**   We use texts from the Project Gutenberg collection (via HuggingFace), which consists of works in the public domain in the United States. In accordance with Project Gutenberg's terms[5], we do not redistribute their format or branding. Our use is strictly limited to non-commercial academic research, in compliance with Swiss copyright law.

**Fineweb-Edu**   We use the Fineweb-Edu corpus released on HuggingFace[6], which is distributed under the Open Data Commons Attribution License (ODC-By) v1.0. Our use complies with this license and adheres to CommonCrawl's Terms of Use. The dataset is used solely for non-commercial academic research, and we do not redistribute the data or any derivatives.

# F  Demonstrations

---

[5]https://www.gutenberg.org/policy/license.html
[6]https://huggingface.co/datasets/HuggingFaceFW/fineweb-edu

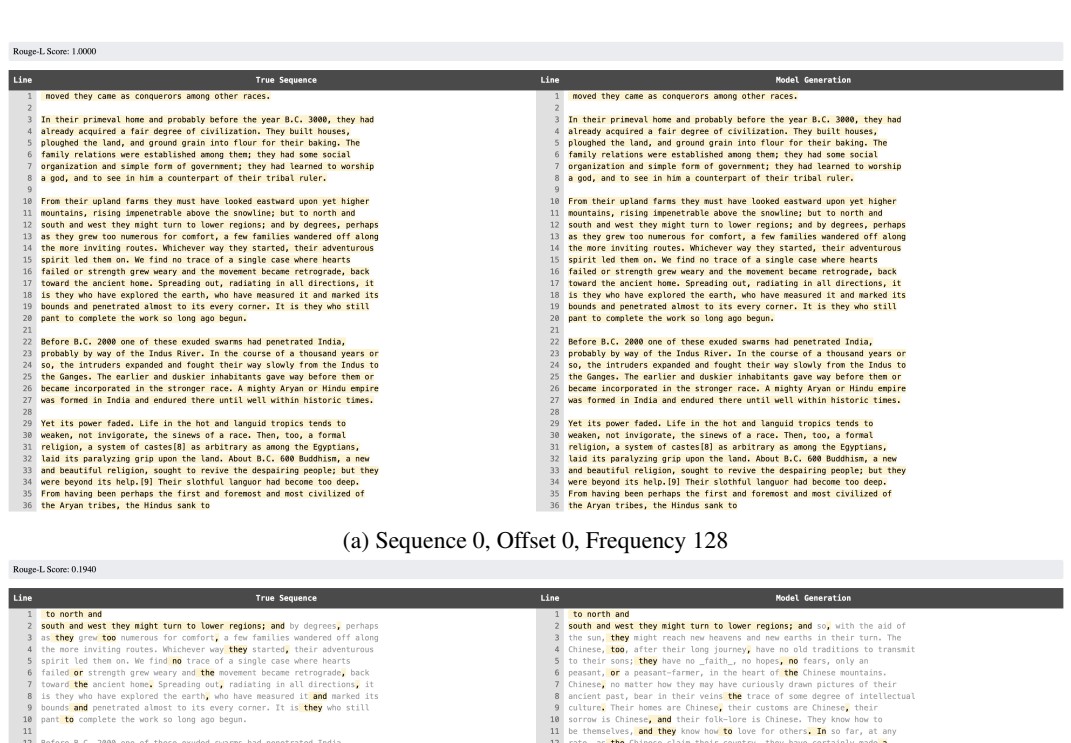

(a) Sequence 0, Offset 0, Frequency 128

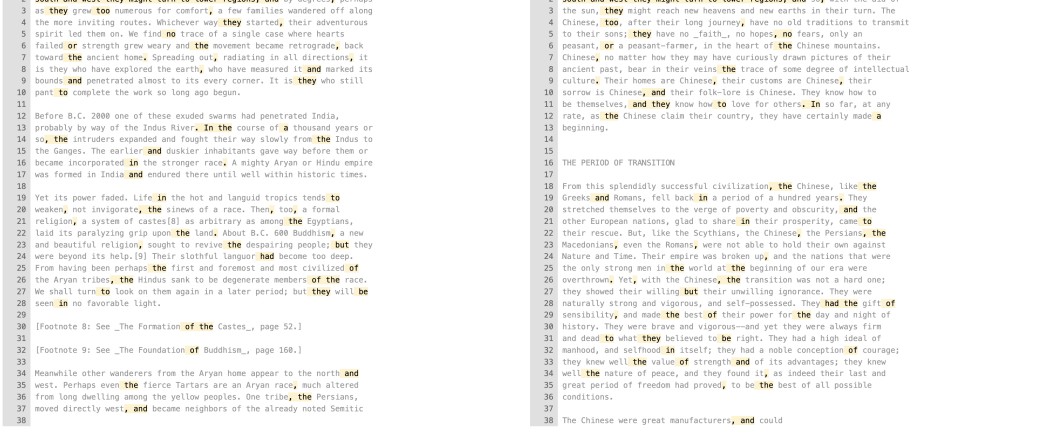

(b) Sequence 0, Offset 128, Frequency 128

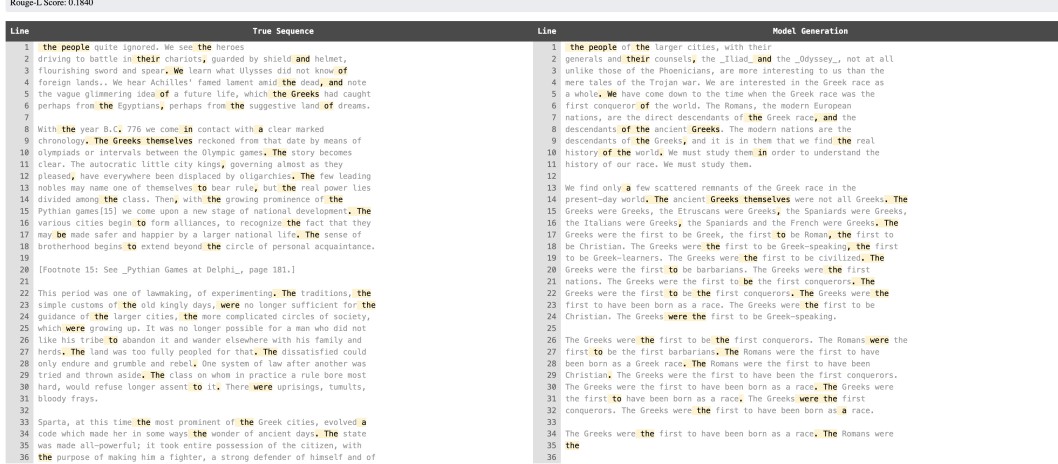

(c) Sequence 0, Offset 2048, Frequency 128

Figure 10: Demonstration of verbatim memorization decay and text degeneration as offset increases in the Sparse Gutenberg setting. Each example shows a 500-token suffix generated from a 500-token prefix by 1B model, with the highlighted span indicating the tokens used for ROUGE-L calculation. At offset 0, the model reproduces the training suffix nearly perfectly. At offset 128, memorization is limited to the initial few tokens. By offset 2048, only scattered fragments are recalled, and the output shows thematic looping—e.g., repetitive structures such as "The xxx were the first to..." emerge in place of coherent continuation.

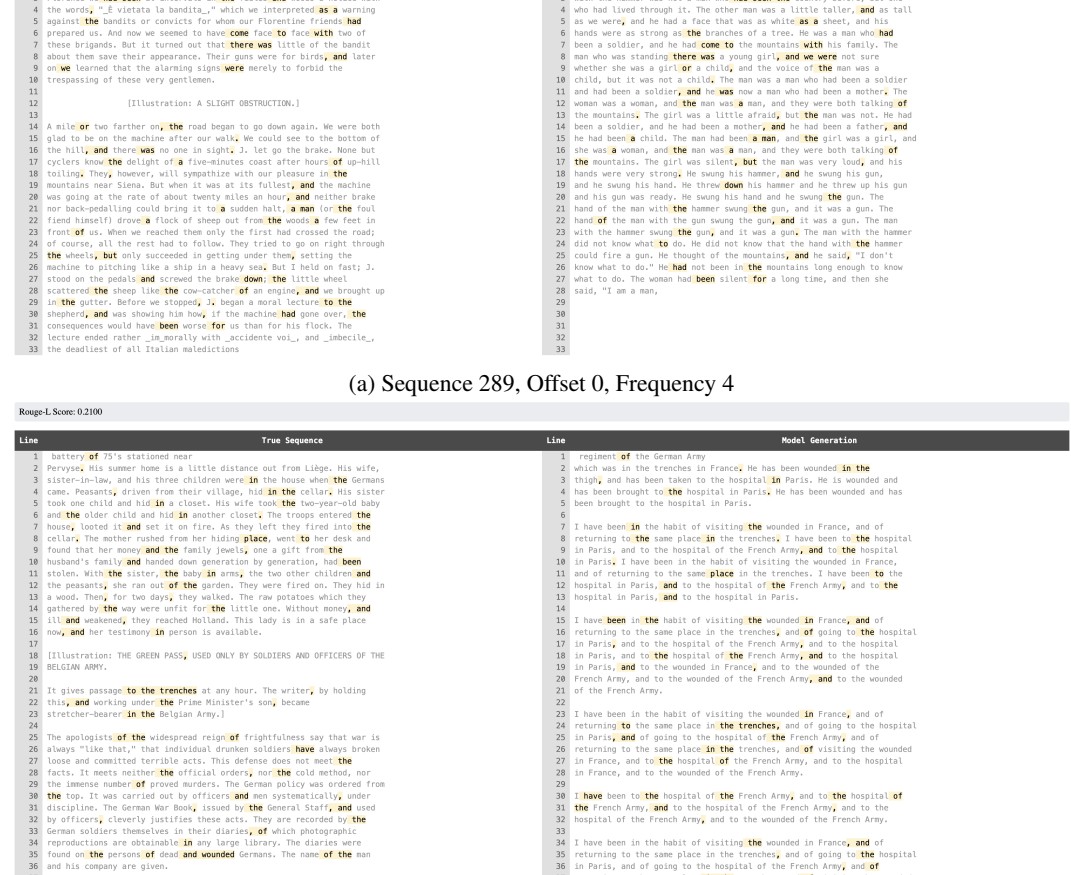

(a) Sequence 289, Offset 0, Frequency 4

(b) Sequence 470, Offset 0, Frequency 4

Figure 11: Demonstration of verbatim memorization decay and text degeneration for low exposure frequency sequences in the Sparse Gutenberg setting. Each example shows a 500-token suffix generated from a 500-token prefix by 1B model, with the highlighted span indicating the tokens used for ROUGE-L calculation. For the generated suffix (right), we show clear evidence of thematic looping and repetitive generations.

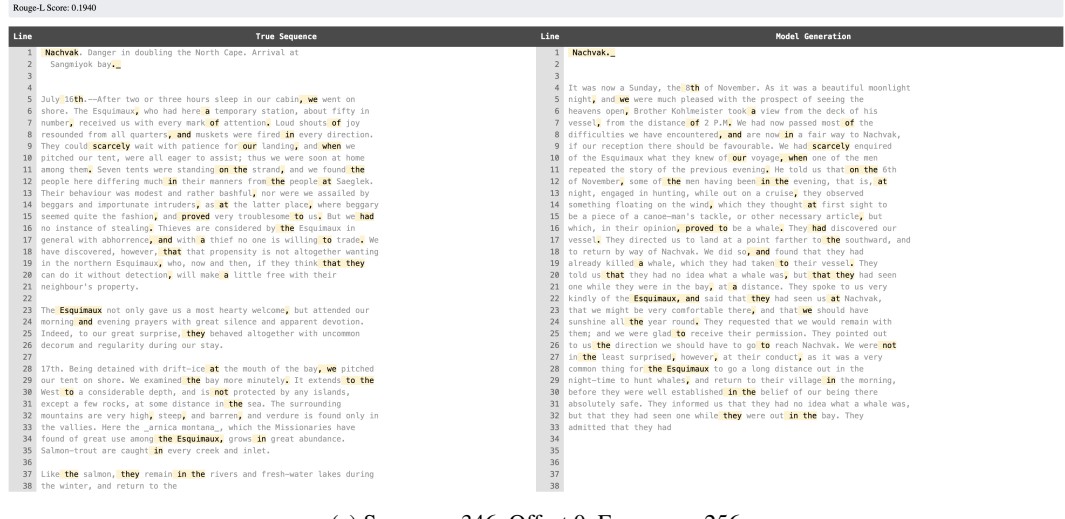

(a) Sequence 346, Offset 0, Frequency 256

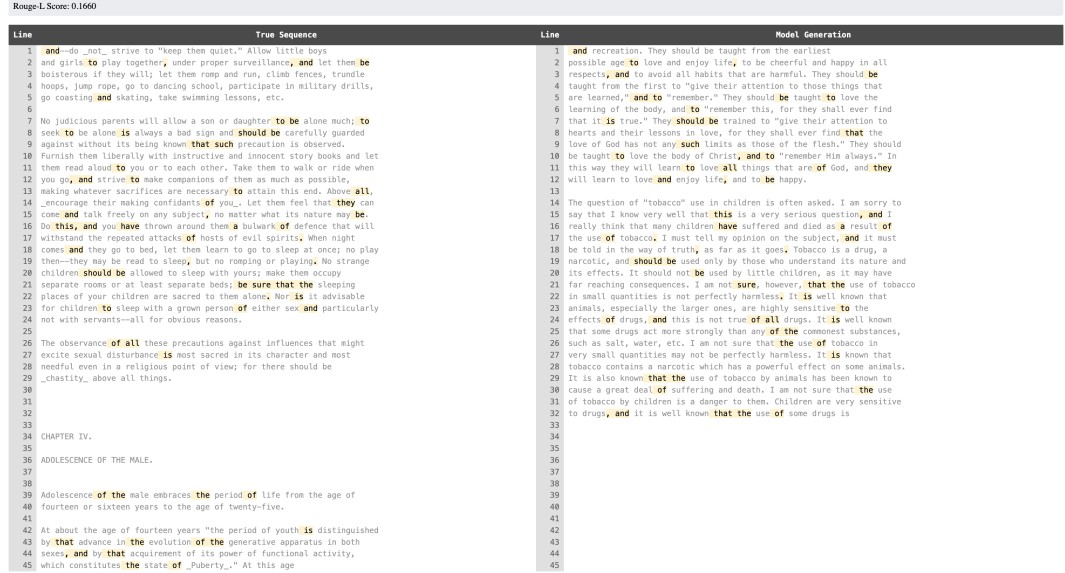

(b) Sequence 440, Offset 0, Frequency 256

Figure 12: Demonstration of how displacing legally sensitive content deeper into the context window effectively suppresses verbatim memorization while preserving semantic fidelity in the Swapped Gutenberg setting. Each example shows a 500-token suffix generated from a 500-token prefix by 1B model, with the highlighted spans indicating portions used for ROUGE-L calculation. (a) Both the true suffix (left) and the model-generated suffix (right) describe interactions and encounters with the Esquimaux community upon arrival at Nachvak, discussing their behavior, hospitality, and local surroundings. Despite thematic similarity, the generated text diverges notably in specific narrative details—such as dates of arrival (July 16th vs. November 8th) and described observations (presence of drift-ice versus whale sightings)—illustrating effective suppression of verbatim memorization with maintained lexical richness and narrative coherence. (b) The true suffix focuses primarily on parenting advice regarding child behavior, education, and guidelines to prevent inappropriate sexual behavior and maintain chastity, whereas the generated suffix addresses the use of tobacco among children and discusses its potential dangers and harmful effects, highlighting clear thematic divergence and effective memorization suppression.

