# OpenReview forum: "Positional Fragility in LLMs: How Offset Effects Reshape Our Understanding of Memorization Risks"
_NeurIPS.cc/2025/Conference — NeurIPS 2025 poster_

### Official Review · Reviewer_Mw7L · 2025-06-20

**Clarity:** 2
**Significance:** 3
**Originality:** 2
**Rating:** 3
**Confidence:** 3

**Summary:**

The paper examines the offset effect for LLM memorization where LLM memorization is taken as text regurgitation from legal perspective.

Built on top of existing works, the paper shows that the text memorization capability depends on where the prefix occurs (so called offset in the paper). More specifically, the memorization rate decreases significantly when an offset is applied (e.g. see Figure 3) where the authors train their models under controlled settings. Based on this observation, the paper proposes further a method that pretrains the LLM with offsets and shows that the memorization rate significantly decreases.

**Questions:**

Can the authors expand on offset definition so that everyone is able to implement it please?

**Ethical Concerns:**

["NO or VERY MINOR ethics concerns only"]

**Final Justification:**

I initially focused on a confusion between perplexity score as performance and the mitigation solution and it has been largely explained in detail in the authors' rebuttal. The authors have also clarified the offset setting which I was not able to get clear during review.

Based on this, I have increased my score. However, I don't think the random prefix mitigation techniques in training time and test time mitigation is a production ready solution and the described techniques currently miss slightly on clarity. So I re-score the paper as slightly below acceptance overall.

**Limitations:**

The limitation section Appendix B does not list significant limitations of the current work. The section is short and discusses that it is empirical, does not address differential privacy and finally a decoding technique coverage. The first one (or even first two) are two general and are not limitations to this particular work; while the last one is a very minor limitation that omitting does not hurt the paper.

Meanwhile arguably there are many limitations to this work (as many of the research works) including testing on popular LLMs, training on more realistic settings, etc. just to name a few.

**Quality:**

2

**Strengths And Weaknesses:**

Strength: the fact that the model is able to regurgitate a large chunk of text is an important phenomenon to be mitigated and the paper has brought very good arguments on it. The paper's main claim (relationship between offset and LLM memorization behaviour) is backed up with controlled experiments.

Weakness:
Firstly, the proposed technology arguably compromises too much performance. As shown in Table 2, the perplexity increases a lot compared to the base case to lower memorization ratio.
Secondly, the offset settings are not clear. The notion is everywhere in the paper however, how offset is setup is only described a bit vaguely in line 68. Unforutunately, I don't know how the skip is implemented concretely from the paper (are they irrelevant tokens, etc.?) I can only guess irrevalent tokens because that is the method adopted in section 6.
After reading the paper, I am not sure if offset simply makes the model perform worse in general (and by consequence have decreased memorization ratio) or the model only worse at memory.
The limitation section has not mentioned any of the above, unfortunately.

---

> ### Author Rebuttal · Authors · 2025-07-31
>
> We sincerely regret that our explanations, particularly regarding the 'offset' setting and perplexity scores, created confusion. We thank the reviewer for highlighting these specific areas, as it gives us an opportunity to improve the paper's clarity for all readers.
>
> &nbsp;
>
> ### Weakness 1: The proposed technology arguably compromises too much performance on perplexity
> ---
> Firstly, we would like to address the concern that our proposed mitigation "compromises too much performance.” The most accurate measure of a model's general capability is its performance on a suite of standard downstream tasks. As shown in Table 5 from Appendix C2 and table 11 from Appendix D3, our 1B and 8B models trained under "*Swapped Gutenberg*" setting achieve highly competitive, and in some cases superior, results across a range of benchmarks when compared to the standard "*Sparse Gutenberg*" models.
>
> &nbsp;
>
> | Model | Wiki. ppl↓ | Hella. norm↑ | ARC-C acc↑ | Wino. acc↑ | CSQA acc↑ | MMLU acc↑ |
> |-------|------------|--------------|------------|------------|-----------|-----------|
> | Standard 1B | **18.71** | **52.31** | **33.36** | 71 | 53.91 | **23.65** |
> | Swapped 1B | 22.31 | 52.22 | 31.83 | **71.93** | **55.01** | 23.27 |
> | Standard 8B | **13.15** | **65.74** | **42.24** | 75.52 | **61.88** | 24.53 |
> | Swapped 8B | 15.71 | 65.48 | 42.06 | **75.79** | 61.09 | **25.89** |
>
> &nbsp;
>
> Secondly, having established that the model's general performance is preserved, we now address the reviewer's point about the perplexity score. We recognize that our table notation (Perplexity↓) may have contributed to confusion, and we appreciate the opportunity to clarify.
>
> While a lower perplexity score typically indicates better predictive performance, its interpretation is inverted when the goal is to **mitigate verbatim memorization**. A model that has perfectly memorized a sequence will exhibit a very low perplexity (approaching 1.0) because it is extremely certain of the next token. Our mitigation's objective is to disrupt this certainty. Therefore, a **higher perplexity** on these specific target sequences is **a positive sign of successful mitigation**, as it shows the model is no longer regurgitating the text. The data for our 8B model clearly demonstrates this phenomenon:
>
> &nbsp;
>
> | Repetition Freq. | Perplexity (Sparse) | Perplexity (Swapped) | Interpretation |
> |------------------|---------------------|----------------------|----------------|
> | 1 | 16.089 | 82.206 | High uncertainty in both cases; difference likely magnified by positional effect. |
> | 8 | 3.43 | 12.841 | Mitigation successfully increases uncertainty as memorization begins. |
> | 64 | 1.023 | 3.67 | Mitigation effectively disrupts strong memorization. |
> | 128 | 1.012 | 3.636 | Mitigation effectively disrupts strong memorization. |
>
> &nbsp;
>
> As shown, for sequences seen 64 or 128 times, the standard model has a near-perfect perplexity score, indicating heavy memorization. Our mitigation strategy successfully increases this perplexity, demonstrating its effectiveness. We also note that the large perplexity difference at a frequency of 1 is likely magnified by a confounding positional effect, as the perplexity is measured on suffixes at different absolute positions:
> - **Base Case (Sparse)**: Suffix position is 500-1000 tokens.
> - **Swapped Case**: Suffix position is 4500-5000 tokens.
>
> &nbsp;
>
> ### Weakness 2: the offset settings are not clear
> ---
> We thank the reviewer for highlighting that the definition and implementation of "offset" were not sufficiently clear. To resolve this, it is crucial to clarify the two distinct ways positional shifts is used in our paper: first as a new evaluation dimension, and second as a training-time mitigation strategy.
>
> **"Offset" as a New Evaluation Dimension**:
> - **Definition**: Given a training text segment from the FM-Probe set, we skip the first `o` tokens (offset `o`) and take the next `n` tokens as the prefix. **No tokens are inserted**; nothing about training changes.
> - **Implementation**: For a document d, an offset of `o=64` means the probing prefix of `n=500` tokens is `d[64:564]`. We then measure verbatim recall on the true following suffix.
>
> **The "Swapped Part" as a Training-Time Mitigation (Section 6)**:
> - **Implementation**: For each Gutenberg sequence, we replace the first 4,000 tokens with unrelated Gutenberg text (randomly sampled), pushing the sensitive portion (the “retained part”) deeper into the context window.
> - **Rationale**: Our main finding is that LLMs are positionally fragile: memorization is easiest when the trigger prefix sits at the very start of the context window. By pushing the sensitive part far from this anchoring zone, we drastically reduce extractable memorization while preserving quality.
>
> Crucially, **offset has nothing to do with the model's general performance**, all downstream performance evaluations reported in our paper (e.g., MMLU, HellaSwag) are conducted using the **standard, un-offsetted evaluation protocols** for those tasks. The strong results on these benchmarks are a true measure of the model's capabilities, independent of our memorization analysis.
>
> &nbsp;
>
> ### Limitations
> ---
> We thank the reviewer for their valuable feedback on our limitations section. We agree that the initial version was not comprehensive enough and did not adequately cover the specific limitations of our work. We will significantly expand and revise this section in the final version of the paper based on the excellent points raised by yourself and other reviewers.
>
> The updated limitations section will include a more thorough discussion of:
> - **Scope as a Proof of Concept (POC)**: We will explicitly state that our work is a foundational study to validate the principle of positional fragility and that our mitigation is a POC, not a production-ready solution. We will clarify that bridging this gap requires both significant engineering and further research, for instance, to develop more granular, controllable memorization mechanisms .
> - **Generalizability of Findings**: We will discuss the need for further research to test if our findings generalize beyond literary texts to other domains, such as PII or medical records .
>
> Last but not least, we agree that testing on popular models in realistic settings is crucial. We would like to clarify that our study was conducted on models from the Llama 3 family, which represent a popular and foundational architecture in the field. Furthermore, to directly address the point about realistic training, we conducted continuous pretraining on a **Llama 3.2 1B pretrained-only** model under our "*Sparse Gutenberg*" setting. The results confirmed that our core findings on positional fragility persist in this more realistic scenario. The full details of this experiment are provided in our response to Reviewer **mkAN**.
>
> &nbsp;
>
> ---
>
> We thank the reviewer again for their time and for providing feedback that will help us improve the paper's clarity. We hope that our detailed responses and new experimental results have fully addressed the initial concerns and provided a clearer perspective on our work's contributions.

---

> ### Author Response · Authors · 2025-08-07
> **Thank You and Follow-up**
>
> Dear Reviewer Mw7L,
>
> Thank you for your time in re-evaluating our manuscript after reading our rebuttal. We sincerely appreciate you raising the scores for the paper's Quality, Significance, and Originality. We are committed to improving the paper as much as possible and would be very grateful for a final piece of guidance.
>
> First, could you please confirm if our rebuttal successfully addressed your main concerns, particularly regarding the model's performance and the definition of the "offset" settings? Second, if there are any other outstanding concerns or limitations that we have not yet addressed, we would sincerely appreciate the opportunity to understand and clarify them. Your feedback is crucial for us to strengthen the final version of our work.
>
> Thank you again for your valuable engagement.
>
> Sincerely,
>
> The Authors of Submission 12369

---

### Official Review · Reviewer_mkAN · 2025-07-04

**Clarity:** 3
**Significance:** 3
**Originality:** 3
**Rating:** 4
**Confidence:** 4

**Summary:**

This paper looks into understanding verbatim memorization in large language models by introducing the concept of "positional fragility" and demonstrating how the position of content within the context window dramatically affects memorization behavior. The authors conducted systematic experiments training LLaMA-style models (1B, 3B, 8B parameters) from scratch to examine memorization patterns, revealing that verbatim memorization is most strongly triggered by short prefixes drawn from the beginning of the context window, with memorization decreasing counterintuitively as prefix length increases.

**Questions:**

See weakness

**Ethical Concerns:**

["NO or VERY MINOR ethics concerns only"]

**Final Justification:**

The reviewer's rebuttal addressed most of my concerns. I keep my positive score unchanged

**Limitations:**

See weakness

**Quality:**

3

**Strengths And Weaknesses:**

**Strengths**
1. Novel finding on the positional offset, memorization drops when content is positioned away from the beginning of the context window. It  challenges prior assumptions about uniform memorization behavior. It is an interesting finding that shifting sensitive sequences deeper into the context window significantly reduces extractability without compromising general performance

2. The paper does very comprehensive analysis, it train LLaMA-style models on 83B tokens from scratch, it gives precise control over training dynamics and data composition. This work also targets at the full-context sequences four times longer and evaluates suffixes at least ten times longer.

3. The "Swapped Gutenberg" approach shows that simply shifting sensitive content 4000 tokens deep into the context window effectively suppresses memorization while maintaining generation quality.

4. The connection between memorization and text degeneration is interesting and important. The observation that models produce repetitive, low-quality output when they cannot retrieve memorized content suggests that memorization and generation quality are more interwined than previously understood.

**Weakness**
1. The model trained on 83B tokens from scratch is very under-trained and curious whether this would impact the model's behaviors on memorization. It would be interesting if the authors could study the same phenomenon by continuing pretraining an existing LMs and compares the difference of memorization behaviors.

2. The paper is the first to show that pretraining language models with smaller batch sizes could greatly reduce the memorization. It would be interesting to do further analysis on why this is the case because this result challenges conventional finding from the differential privacy literature, where larger batch sizes typically diminish the influence of individual examples and are generally preferred for privacy preservation.

3. The Swapped Gutenberg is an effective approach but it requires encoding of an long input prefix and does the cross attention. This may not be very practical to deploy in the real world. It would be interesting to further quantify the potential side effect of Swapped Gutenberg in terms of downstream task performance. The long prefix may distract the model from using the useful knowledge encoded in the model weights. One related pointer is from Gemini 2.5 report discussing that when the context goes too long, the model is stuck in the context poisoning problem, hard to decide when to generate new information and retrieve information in the existing context.

4. It would also be interesting to delve deeper into how the swapped part affects generation. Is memorization reduced because the model has a tendency to copy directly from the swapped part?

---

> ### Author Rebuttal · Authors · 2025-07-31
>
> We sincerely thank the reviewer for their positive and highly constructive feedback. In response to the insightful questions raised, we have conducted several new experiments and analyses, which we summarize below.
>
> &nbsp;
>
> ### On the "Under-trained" Concern
> ---
> The reviewer notes that our models, trained on 83B tokens, are not trained to the same scale as today's largest foundation models. This was a deliberate and principled choice for the following reasons:
> - While state-of-the-art models are trained on trillions of tokens, the renowned "Chinchilla" scaling laws suggest a compute-optimal parameter-to-token ratio of approximately 1:20 [1]. Our 1B and 3B models, trained on 83B tokens, are well above this threshold.
> - To maintain consistency across our experiments and work within a feasible academic budget, we applied the same 83B token count to all models, including our 8B model (a 1:10 ratio). This ensures a fair and controlled comparison of model scale effects.
>
> We agree with the reviewer that studying our findings in a continued pretraining paradigm is a valuable test of their generalizability. To this end, we conducted a new set of experiments with the following setup:
> - **Base Model**: Llama 3.2 1B pretrained-only model (9T tokens).
> - **Training**: continued pretraining on the "Sparse Gutenberg" dataset using otherwise identical settings, but with a learning rate that decayed from 5e-5 to 1e-5.
> - **Comparison**: We compare this "Continue" model directly against our original 1B model "trained from scratch." Both models were evaluated using 500-token prefixes to generate 500-token suffixes.
>
> **Our new experiments confirm that our central findings are robust and hold true across both training paradigms**:
> 1. **Positional Fragility Persists**: The "Continue" model exhibits the same dramatic drop in verbatim recall (`Rouge-L`) as the offset increases, confirming this is a fundamental behavior .
> 2. **The Link Between Memorization and Degeneration Holds**: At low repetition counts where memorization fails, the "Continue" model also exhibits significant text degeneration. Interestingly, its degeneration is even more pronounced (`TTR_gen` of ~0.17 vs. ~0.23 for the "Scratch" model), suggesting the link is stronger in more capable models .
> 3. Additionally, we found that the 1B 'Continue' model did not outperform our 8B model trained from scratch on downstream tasks.
>
> | Offset | Rouge-L || Ref_PPL || TTR_gen ||
> |--------|---------|----------|----|------|---------|-----|
> |        | Scratch | Continue | Scratch | Continue | Scratch | Continue |
> | 0      | 0.744   | 0.669    | 1.051   | 1.029    | 0.521   | 0.521    |
> | 4      | 0.524   | 0.596    | 1.072   | 1.036    | 0.509   | 0.512    |
> | 8      | 0.463   | 0.569    | 1.079   | 1.048    | 0.491   | 0.498    |
> | 32     | 0.346   | 0.426    | 1.127   | 1.092    | 0.465   | 0.482    |
> | 128    | 0.219   | 0.249    | 1.356   | 1.283    | 0.411   | 0.437    |
> | 512    | 0.191   | 0.202    | 2.109   | 1.761    | 0.345   | 0.364    |
> | 2048   | 0.185   | 0.188    | 3.305   | 2.265    | 0.293   | 0.340    |
>
> | Repetitions | Rouge-L || Ref_PPL || TTR_gen ||
> |-------------|---------|----------|---------|----------|---------|-------|
> |             | Scratch | Continue | Scratch | Continue | Scratch | Continue |
> | 1           | 0.181   | 0.178    | 28.957  | 17.302   | 0.225   | 0.178    |
> | 4           | 0.184   | 0.178    | 21.057  | 14.252   | 0.228   | 0.160    |
> | 8           | 0.183   | 0.180    | 14.202  | 11.404   | 0.231   | 0.171    |
> | 16          | 0.183   | 0.182    | 6.886   | 6.804    | 0.323   | 0.293    |
> | 64          | 0.522   | 0.446    | 1.130   | 1.078    | 0.497   | 0.503    |
> | 128         | 0.744   | 0.669    | 1.051   | 1.029    | 0.521   | 0.521    |
>
> In summary, these new results strongly support our original conclusions. The phenomenon of positional fragility is not an artifact of our models being "under-trained" but appears to be a general behavior that persists even when continuing the pretraining of an existing, state-of-the-art language model.
>
> &nbsp;
>
> ### On Batch Size and Memorization
> ---
> We thank the reviewer for raising this important point. Below, we clarify the distinction between DP and standard language model training, and provide a plausible explanation—based on exposure frequency and training stochasticity—for the observed reduction in memorization with smaller batch sizes.
>
> **The Unique Dynamics of Differentially Private Training**
>
> In the DP setting, Differentially Private Stochastic Gradient Descent (DP-SGD) clips per-sample gradients and adds noise to ensure privacy guarantees. Larger batch sizes improve the signal-to-noise ratio, mitigating the utility loss caused by added noise, and are therefore often preferred for a favorable privacy-utility trade-off [2].
>
> In contrast, standard non-DP language model pretraining (as in our work) does not add noise. As noted in Appendix D.2, prior studies have shown that smaller batch sizes and longer training durations often improve generalization, consistent with our observed reduction in memorization.
>
> **Exposure Frequency**
>
> Given a fixed compute budget and a constant total number of tokens processed, batch size and the number of gradient updates are inversely proportional. In an ideal case, for the 128-repetition FM-probe bucket, our experimental data from Table 7 (Appendix D.2) reveals a stark contrast: with a small global batch size (GBS) of 60, the model performs approximately 1,328 gradient updates (170,005 ÷ 128) between each exposure to the same sequence. With a large GBS of 240, it performs only about 332 updates (42,501 ÷ 128).
>
> This fourfold difference in update frequency suggests a direct mechanism for increased memorization. With larger batches, the model's parameters have significantly less opportunity to be altered by other data between repeated exposures, which may reinforce and accelerate the memorization of that specific sequence. Conversely, the numerous updates in the small-batch setting likely introduce sufficient stochasticity to disrupt the formation of verbatim memory.
>
> &nbsp;
>
> ### On Practicality and Side Effects
> ---
> We thank the reviewer for this insightful question regarding the practicality and potential side effects of the "Swapped Gutenberg" mitigation, such as performance degradation or context poisoning.
>
> **On Practical Deployability**: While our "*Swapped Gutenberg*" mitigation is intended to be a Proof of Concept (POC) , we have already outlined a feasible path to its implementation. As detailed in our response to Reviewer **vJB5**.
>
> **On Downstream Performance**: Regarding the potential for the long prefix to harm downstream performance, our evaluation results show this is not the case, as detailed with a comparative table in our response to Reviewer **Mw7L**.
>
> **On Context Poisoning**: With the model's general performance preserved, we now clarify why the "context poisoning" phenomenon is not directly applicable to our findings. There are two key distinctions between our experimental setup and the settings where this is typically observed:
> - Different Setting (Pre-training vs. Agentic Use): The "context poisoning" issue noted in reports like Gemini 2.5 occurred in an agentic setting, where a model uses its context as a memory for multi-step reasoning. In contrast, our work studies fundamental memorization dynamics that occur during the pre-training stage itself [3] .
> - Different Context Length (Native vs. Extrapolated): These distraction effects are generally documented when models are pushed to lengths far exceeding their pre-training window (e.g., 100k+ tokens). Our experiments operate entirely within the model's native 8,192-token context window.
>
> Therefore, while the concern is valid for ultra-long context models, it is not directly applicable to the findings presented in our work.
>
> &nbsp;
>
> ### On Copying from the "Swapped Part"
> ---
> This is a critical alternative hypothesis that warrants a direct empirical investigation. To address this, we conducted a thorough contamination analysis, complemented by the qualitative examples provided in our appendix.
>
> Since our work focuses on measuring verbatim memorization, the most direct method to test this hypothesis is to quantify the verbatim overlap between the model's output and the swapped prefix. We performed a 13-gram contamination analysis with the following setup:
> - Model: Pretrained 8B model under "*Swapped Gutenberg*" setting.
> - Inference Task: To test the strongest possible version of the "copying" hypothesis, the model was prompted with 500-token prefixes from the Gutenberg dataset (the "retained part") placed at a 0-token offset, immediately following the “swapped part”. It was then tasked with generating a 500-token suffix.
> - Contamination Check: For the generated suffixes of each FM-probe bucket, we compare them against the entirety of their corresponding 4000-token "swapped content” from the FineWeb-edu dataset to identify any matching 13-grams.
>
>  The analysis shows contamination is negligible, peaking at just **0.04%** . This quantitative result, supported by qualitative examples in Figure 12 , confirms the model is not copying. The mitigation works by disrupting the stable positional context required for memory retrieval, reinforcing our central thesis of positional fragility .
>
> &nbsp;
>
> ---
> We once again thank the reviewer for their highly constructive and thought-provoking questions. We hope this detailed response has clarified the deeper implications of our work.
>
> **References**:
>
> [1] Jordan Hoffmann, et al., "Training Compute-Optimal Large Language Models," arXiv preprint, 2022.
>
> [2] Martin Abadi, et al., "Deep Learning with Differential Privacy," Proceedings of the 2016 ACM SIGSAC CCS, pp. 308-318, 2016.
>
> [3] Gheorghe Comanici, et al., "Gemini 2.5: Pushing the Frontier with Advanced Reasoning, Multimodality, Long Context, and Next Generation Agentic Capabilities," arXiv preprint, 2025.

---

> ### Comment · Reviewer_mkAN · 2025-08-01
>
> Thank you for the detailed response. It addressed most of my concerns

---

> ### Author Response · Authors · 2025-08-03
> **Thank You and Follow-up**
>
> Dear Reviewer mKAn,
>
> Thank you so much for your positive feedback and for confirming that our rebuttal has addressed most of your concerns. We truly appreciate your constructive engagement.
>
> To help us fully resolve everything, we were hoping you might elaborate on any minor points that may still be outstanding. We are eager to use the remainder of the discussion period to ensure our work is as clear and convincing as possible for you.
>
> Thank you again for your guidance.
>
> Sincerely,
> The Authors of Paper 12369

---

### Official Review · Reviewer_vjB5 · 2025-07-05

**Clarity:** 3
**Significance:** 2
**Originality:** 3
**Rating:** 5
**Confidence:** 3

**Summary:**

This paper analyzes the memorization in LLMs, with an emphasis on sensitive data memorization such as copyright-protected data. The authors introduce a new aspect to be considered in memorization analysis, offset effect, and identify new phenomena, positional fragility---memorization is significantly more likely when the probing prefix appears near the start of the model’s context window, and degrades sharply when the prefix is offset deeper into the sequence. To study this, the authors pretrain LLaMA-style models (1B, 3B, 8B) on 83B tokens using a mix of FineWeb-Edu and Project Gutenberg, with careful control over exposure frequencies and token positions. They introduce the Swapped Gutenberg experiment, where the first 4K tokens of each training sequence are replaced with random text to push potentially sensitive content deeper into the sequence. This reduces memorization without compromising output quality.

**Questions:**

1. How do you envision operationalizing the "Swapped Gutenberg" mitigation in large-scale pertaining?
2. In your setup, Project Gutenberg is used as the sensitive subset - what motivated this choice, and can the experiments be applied to other domains or setups such as PII or other sensitive data domains such as medical domains?

**Ethical Concerns:**

["NO or VERY MINOR ethics concerns only"]

**Final Justification:**

I'm increasing my score as I believe the authors' response addressed my concerns.

**Limitations:**

Yes

**Quality:**

3

**Strengths And Weaknesses:**

**Strengths**
- This paper introduces a novel axis for analyzing memorization in LLMs: the position of a prefix within the context window during training and evaluation.

- The experiments are rigorous and comprehensive, with strong empirical support across model scales and memorization frequencies. The authors pretrain LLaMA-style models (1B, 3B, and 8B) on an 83B-token corpus specifically for this study.

- The finding that failures in verbatim memorization can lead to output degeneration adds a valuable dimension to existing work on repetition-induced degeneration.

- The authors propose a simple but effective memorization mitigation strategy, displacing sensitive sequences later in the context window during training. This method significantly reduces extractable memorization without compromising generation quality.

**Weaknesses**

1. Limited practical utility of findings

While the findings are novel and empirically supported, the practical application of the mitigation strategy remains unclear. Specifically:

- The authors propose shifting sensitive content deeper into the context window (Swapped Gutenberg setup), but do not discuss how such a strategy could be applied during LLM pretraining, especially in large-scale, distributed training pipelines.
- In real-world scenarios, training data is rarely labeled for sensitivity, and existing pretraining frameworks do not typically support fine-grained control over token placement or prefix location during batching.
- The paper would benefit from a brief discussion of the feasibility of operationalizing this technique; for instance, through modified data loaders, selective shuffling, or integration with deduplication or data curation workflows.

Additionally, while I appreciate the authors’ efforts to pretrain models specifically for this study, applying this analysis to existing models is difficult in practice. For example, models like LLaMA do not release their pretraining data, and even open-source efforts like OLMo do not provide detailed logs (e.g., which sequences appear in which batches), making it nearly impossible to retroactively analyze prefix positions or apply these findings directly to such models.

2. Unrealistic deployment assumptions
The mitigation strategy seems to assume that:
- The model developer knows which parts of the training data are sensitive.
- The developer has control over where in the sequence those parts are placed.

In practice, especially with trillion-token-scale training, we often lack precise metadata on content sensitivity, and restructuring input sequences at that scale would require non-trivial changes to data pipelines.

Overall, while this paper contributes a compelling new perspective, the evaluation is conducted in a highly controlled, clean experimental setting, where Project Gutenberg is treated as the sensitive subset and FineWeb as safe. The analysis also relies on precise knowledge of where a prefix appears during training and evaluation, which may not be realistic in most practical scenarios. These assumptions limit the generalizability of the proposed analysis and mitigation to real-world LLM training.

---

> ### Author Rebuttal · Authors · 2025-07-31
>
> We thank the reviewer for their insightful and detailed feedback. The questions raised about practical utility and operationalization are critical for bridging the gap between scientific discovery and large-scale engineering. We value this opportunity to clarify the precise nature and intended contribution of our work.
>
> We acknowledge that our phrasing, such as the term "*proposed mitigation strategy*" in line 225 and 233, may have created ambiguity. To clarify, our work's primary contribution is that of a **foundational study**, and the "*Swapped Gutenberg*" experiment should be understood as a **Proof of Concept (POC)**. Our intent was to empirically validate a previously overlooked phenomenon—the positional fragility of memorization—and our POC serves to demonstrate that this core finding can be leveraged for mitigation. This was a deliberate choice in how we framed our contribution; for instance, our abstract and introduction focus on "*leveraging this finding*", rather than formally proposing a complete, operationalized method.
>
> By establishing this principle, we provide the critical justification for the community to now invest in solving the associated engineering challenges. With this framing in mind, we address the specific concerns below.
>
> &nbsp;
>
> ## On "Unrealistic Deployment Assumptions" and Feasibility (Weakness 1)
> ---
> The reviewer's critique is centered on two key assumptions they believe to be unrealistic for large-scale LLM training. We address each directly, arguing that both are consistent with state-of-the-art LLM development.
>
> &nbsp;
>
> ### Assumption 1: "The model developer knows which parts of the training data are sensitive."
>
> We argue this assumption is realistic, as identifying and tracking sensitive data is now both a practical necessity for developers driven by legal and ethical imperatives and a technically maturing engineering field.
>
> **Legal and Commercial Necessity**: It is no secret that proprietary models are trained with copyrighted texts. Recent legal rulings, such as the landmark Bartz v. Anthropic (2025), have established that training on such works can be considered fair use, but this defense critically hinges on the model not producing infringing outputs that could harm the market for the original work. To manage this immense legal and financial risk, a developer must be aware of which portions of their training data consist of copyrighted materials.
>
> **Sensitive Data Identification**:
> -  For amorphous types of sensitive data, e.g. PII and toxicity, tracing this content at scale is a mature engineering field. Modern data curation pipelines like NVIDIA's NeMo Curator  and Hugging Face's DataTrove are designed for this purpose, and a "sensitivity detection" module can be implemented within these frameworks using established methods.
> - For copyrighted content, the problem is even more tractable. The most robust and efficient approach is to tag content with its source and copyright status at the point of ingestion, as it is often sourced from discrete works like books and articles.
> - Even if not tagged at the source, open-source research tools like DE-COP [1] and commercial APIs from services like Copyleaks and Patronus AI's CopyrightCatcher can be integrated directly into these pipelines to flag protected works.
>
> **Reproducibility by Design**: The premise that retroactive analysis is "nearly impossible" is inconsistent with the design of modern frameworks. Frameworks like Megatron-LM use an `IndexedDataset` format (`.bin`/`.idx` files) that creates a permanent mapping from any training sample back to its source document. The data loader then uses a global random seed to generate a deterministic and cached  `shuffle_index`, which dictates the exact order of data loading for the entire run. This means the training metadata is, in fact, logged, making retroactive analysis entirely possible.
>
> **The Emerging Reality of Traceability**: The technical obstacles to retroactive analysis are also being actively dismantled. Recent tools such as OLMoTrace demonstrate that it is now possible to trace model outputs back to specific training samples in multi-trillion-token datasets in real-time [2].
>
> &nbsp;
>
> ### Assumption 2: "The developer has control over where in the sequence those parts are placed."
>
> This is a tractable offline data preparation task, not a complex modification to the distributed training loop itself. Modern pretraining frameworks explicitly separates data preparation from the training process, providing a clear point of intervention.
>
> **Offline Data Preparation vs. Online Training**: In standard LLM workflows, the corpus is pre-processed, tokenized, and packed into sequences in an offline stage. Our "*Swapped Gutenberg*", or any sequence restructuring is an intervention in this simpler, more manageable offline stage. The distributed training engine (e.g., Megatron-LM, DeepSpeed) is agnostic to how sequences were constructed; it simply fetch the pre-tokenized data.
>
> **Feasibility via Custom Offline Data Logic**: The manipulation is achieved by creating custom logic within the offline data preparation scripts that process raw text into a final, tokenized training asset. The engineering precedent for intelligently organizing documents is already established by techniques like "best-fit packing" and "dataset decomposition," which improve training efficiency by moving beyond naive concatenation. The operationalization the "*Swapped Gutenberg*" mitigation can simply leverage pre-existing sensitivity labels to inform this sequence construction.
>
> Therefore, having precise metadata on content sensitivity is not an unrealistic ideal; it is a fundamental component of **responsible AI**.
>
> &nbsp;
>
> ## On "Limited Practical Utility" and Generalizability (Weakness 2)
> ---
> Having established feasibility, we now address practical utility. While applying our analysis to closed models is difficult, this is a symptom of the field's lack of transparency, not a weakness of our method. Our work's true utility lies in unlocking new capabilities for **auditing** and **advocacy**, providing the scientific basis to demand such transparency.
>
> &nbsp;
>
> ### A New Paradigm for Auditing: The "Sliding Window" Attack
>
> Our findings enable a new, more powerful auditing technique: the **"slide window" attack**. Our research has profound implications for AI security, particularly for Membership Inference Attacks (MIAs). Current MIAs on LLM pre-training data often barely outperform random guessing, and our findings offer a potential explanation for this weakness.
>
> We show that a training example may be a true member of the dataset, yet fail to trigger a memorization signal if the probe does not align with the early tokens in the context window. This leads to a high rate of false negatives in current MIAs, where true members are incorrectly classified as non-members simply because the attack was not conducted in a positionally-aware manner.
>
> The sliding window attack resolves this: an auditor can probe the model at multiple offsets and look for **sharp, localized drops in perplexity** or spikes in `ROUGE`. These “fragility signatures” are stronger evidence of memorization than any single-offset metric, enabling a more precise and positionally-aware audit.
>
> &nbsp;
>
> ### Empowering Stakeholders and Advocating for Transparency
>
> Our research provides stakeholders with a new, powerful tool for advocacy. We show that how content is used during training—specifically, its position within the context window—has a material impact on the resulting model's behavior. Armed with this knowledge, rights holders are no longer limited to asking if their work was used. They can now use our findings to demand greater transparency regarding how it was used, asking for specific training metadata like exposure counts and context window placement. This provides the empirical grounding for policymakers and the public to demand that developers release the crucial training-time metadata necessary for true auditing and reproducibility.
>
> &nbsp;
>
> ## Questions:
> ---
>
> &nbsp;
>
> 1. **Operationalizing the "Swapped Gutenberg" Mitigation**: Detailed in weakness section.
>
> 2. **Why Project Gutenberg was Used**:
>    - **Realistic Simulation**: This research is framed from a legal perspective, focusing on the risks of verbatim memorization of copyrighted works like books and articles, whereas Project Gutenberg consists of long-form literary texts that structurally mimic high-risk copyrighted books, providing a legally relevant testbed for studying copyright issues without actual infringement.
>    - **Experimental Control**: Our contamination analysis confirmed that the Gutenberg texts had negligible 13-gram overlap (**0.34%**) with the general purpose fineweb-edu data, ensuring that any observed memorization could be cleanly attributed to the experiment's controlled factors.
>
> 3. **Application to PII and Medical Data**: While the principle of positional fragility is expected to generalize, applying it to new domains like PII and medical records requires further research and domain-specific adjustments. We believe that with careful calibration, extending this approach to these sensitive domains is feasible.
>
> &nbsp;
>
> ---
> We thank the reviewer again for their valuable feedback. In response to their comments, we will rephrase key sections of the manuscript to better clarify the foundational nature of our study and the role of our mitigation strategy as a Proof of Concept, we will also incorporate our contamination analysis results. We hope these targeted revisions and additions fully address the reviewer's concerns.
>
> **References**
>
> [1] André V. Duarte, et al., "DE-COP: Detecting Copyrighted Content in Language Models Training Data," Proceedings of the 41st International Conference on Machine Learning (ICML), 2024.
>
> [2] Jiacheng Liu, et al., "OLMoTrace: Tracing Language Model Outputs Back to Trillions of Training Tokens," arXiv preprint arXiv:2504.07096, 2025.

---

> > ### Comment · Reviewer_vjB5 · 2025-08-07
> > **Thank you for your comment**
> >
> > Thank you so much for your detailed comment!
> >
> > Could you point me to specific references for open models and trillion-token-scale pretraining corpora where (1) the exact data loading order and indices used during training are publicly available, and (2) high-risk documents (e.g., copyrighted content) are explicitly tagged? While I fully agree that these capabilities are essential for responsible AI, my original comment was focused on the realism of these assumptions in current modeling practice, rather than their theoretical feasibility. I reviewed Megatron-LM but couldn’t find details on their training data composition, sensitivity tagging, or data loading order for released models.
> >
> > I don't dispute that systems can be designed with full traceability and tagging. However, in practice, even among the most open models, such as OLMo (at least in the OLMo-1 release), tracking the exact batch order and loading sequence has proven difficult, particularly when non-sequential data loaders or randomized shuffling are used. In our own prior work, we attempted similar tracing and were unable to reconstruct sample-level orderings. These implementation details are rarely standardized or exposed by current frameworks.
> >
> > Regarding the distinction between offline preprocessing and online training: while it’s true that corpora are typically preprocessed and tokenized offline, I disagree with the notion that maintaining original sequence boundaries or disabling shuffling is standard. It is common to concatenate samples up to the sequence length limit (which can be changed during different training runs, especially in earlier stages) and apply random shuffling for efficiency or regularization. This makes exact positional control difficult in practice.
> >
> > While post-hoc techniques like OLMoTrace are promising, such methods rely on full access to training data—which is rare among strong open-weight models such as Qwen, DeepSeek or Kimi—and as your paper and others point out, access of the original corpus alone is insufficient: understanding how data was ordered, duplicated, or weighted during training is crucial for studying memorization and mitigation strategies. So, having those techniques alone still do not enable this type of analysis.
> >
> > Again, I appreciate the contribution of this work and the authors' detailed response! So while I agree that this work highlights an important direction for building more auditable and accountable systems, I continue to see a gap between the assumptions required for your mitigation and today’s prevailing practices.

---

> ### Comment · Area_Chair_kNU1 · 2025-08-07
> **Please respond to authors.**
>
> Dear Reviewer,
>
> Your active participation in the review process is crucial. Please respond to the authors' response and acknowledge you have done so.
>
> Thanks.
>
> -AC

---

> ### Author Response · Authors · 2025-08-07
> **On the Reproducibility of Data Loading Order in Modern Frameworks (1/2)**
>
> We thank the reviewer for their comment regarding the practical difficulty of tracing data loading sequences in current pretraining frameworks. While it is true that many frameworks obscure these details, we wish to clarify that this is not a fundamental limitation. State-of-the-art open frameworks like AI2's OLMo and NVIDIA's Megatron-LM are explicitly designed for reproducibility, providing different **implementation strategies** to reconstruct the exact data loading order, even when using randomized shuffling.
>
> &nbsp;
>
> ### Case Study: OLMo's Deterministic, Chunk-Level Traceability
> ---
> Thank you for raising OLMo as an example. After examining the OLMo codebase in detail, I can provide technical clarification on what is and isn't possible with their system:
> 1. **Preprocessing and Chunking**:
>    - OLMo uses the Dolma toolkit to tokenize and concatenate documents with `EOS` tokens, then splits them into 2048-token chunks stored in `.npy` files. However, this "concatenate and chunk" strategy means original document boundaries are not explicitly preserved.
>    - e.g.  `[Doc1: 1500 tokens][EOS][Doc2: 3000 tokens][EOS] → [Chunk0: 2048][Chunk1: 2048][Chunk2: 406]`
> 2. **Deterministic Shuffling**: After preprocessing, OLMo implements deterministic data ordering through its `IterableDataset` class (`olmo/data/iterable_dataset.py:91-114`). It uses a seeded generator (`numpy.random.PCG64(seed + epoch)`) and saves the exact shuffled order of chunks to a `global_indices.npy` file. Thus, the exact order is both deterministic and explicitly logged, addressing the concern about non-sequential data loaders.
> 3. **Per-batch Indices Reconstruction**
>    ```python
>    # From OLMo-1B.yaml config
>    seed = 6198
>    global_batch_size = 2048
>
>    # Deterministic shuffle (NOT sequential!)
>    rng = np.random.Generator(np.random.PCG64(seed=seed + epoch))
>    indices = np.arange(dataset_size, dtype=np.uint32)  # [0, 1, 2, ..., N-1]
>    rng.shuffle(indices)  # Now shuffled: e.g., [896398, 797198, 937343, ...]
>
>    # The global_indices.npy file stores this exact shuffled order
>    # So even though it's "randomized", we know sample 896398 was seen first,
>    # sample 797198 second, etc.
>
>    # Batch at step N uses the shuffled order
>    batch_indices = indices[step * global_batch_size:(step + 1) * global_batch_size]
>    ```
>
> &nbsp;
>
> **The Trade-Off: Chunk vs. Document-Level Indexing**
>
> OLMo guarantees reproducibility at the chunk level, an $O(1)$ operation via the saved index files. However, mapping these chunks back to their original source documents is non-trivial. Because document boundaries are implicit (marked only by `EOS` tokens), identifying which documents are present in a given batch requires an $O(n)$ scan of the tokenized data to find those boundaries. Explicit, document-level indexing is lost in this design.
>
> &nbsp;
>
> ### Megatron-LM: Achieving Document-Level Traceability
> ---
> Nvidia's Megatron-LM framework offer a different solution that addresses OLMo's primary limitation. Megatron-LM utilizes an `IndexedDataset` format, which consists of two distinct files:
> - Binary File (`.bin`): This file contains the raw, concatenated token data as `int32` values, with no separators.
> - Index File (`.idx`): This file provides the crucial metadata for document-level indexing, consists of:
>    - Header (34 bytes):
>       - 9 bytes: Magic string `MMIDIDX\x00\x00`
>       - 8 bytes: Version (uint64, always 1)
>       - 1 byte: Data type code (4=int32, 8=uint16)
>       - 8 bytes: Number of sequences
>       - 8 bytes: Number of documents
>    - Data Arrays:
>       - Document lengths: #docs × 4 bytes
>       - Document pointers: #docs × 8 bytes
>       - Document indices: (#docs + 1) × 8 bytes
>
> This `.idx` file explicitly stores the size and offset for each document, creating a persistent, direct mapping from any training sample back to its source document. This approach enables `O(1)` lookup of the document containing any given token. , and the deterministic random shuffled index is also logged, as shown in the implementation (`megatron/core/datasets/gpt_dataset.py#L341`).
>
> &nbsp;
>
> ---
> In conclusion, We agree with the notion that practices like concatenating documents and applying randomized shuffling are common, and **we have never claimed that disabling these features is standard practice**. In fact, the ability to reproduce the exact data loading order from a fixed global seed is essential for robust large-scale training. **Without this capability, it would be impossible to reliably resume training from the frequent interruptions expected during pretraining**. Therefore, this deterministic reproducibility is a well-established and vital component of modern training frameworks. Making the underlying mechanisms and artifacts accessible, however, is a design choice. Modern NLP frameworks, including OLMo's PyTorch implementation and NVIDIA's Megatron-LM, both support this feature, underscoring that the gap is one of accessibility, not technical feasibility.

---

> ### Author Response · Authors · 2025-08-08
> **On the Gap Between Prevailing Practice and Technical Feasibility (2/2)**
>
> We thank the reviewer for crystallizing the core of the issue: the gap between the "current modeling practice" and what is "theoretically feasible."  Our previous response demonstrated that, implementation-wise, there is no obstacle to revealing the exact training sequence ordering. Here, we would like to address the more high-level concerns you have raised.
>
> &nbsp;
>
> ### The Current Landscape of Data Transparency
> ---
> You asked for specific references for trillion-token-scale models where (1) the exact data loading order is publicly available, and (2) high-risk documents are explicitly tagged.
>
> Your questioning here is well-founded, and we will answer directly: to our knowledge, **no** major open model release currently provides both of these artifacts in a simple, downloadable format. This reality validates your point about prevailing practices. However, the full picture is more nuanced:
> - **On Data Loading Order**: The ability to reconstruct the training order is not a missing feature but a matter of effort. As you noted with OLMo, it requires dedicated researchers to re-tokenize the exact data mixture with the same tokenizer and run at least a dry run of the training script with the correct configuration files (e.g., global seed) to generate the global indices. While OLMo release all the necessary components (code, configs, seed) to make this possible, they do not typically provide the final `global_indices.npy` file as a direct download.
> - **On High-Risk Tagging**: Rather than explicit copyright tagging, the current rule of thumb is license-aware data curation. Datasets like **The Stack** are built by filtering for permissively-licensed source code, a practice that has become a standard for responsible data sourcing. However, there are established methods for tagging other types of sensitive content using industrial-strength open-source toolkits like **NVIDIA NeMo Curator** and **HuggingFace DataTrove** with modules for identifying and filtering PII or toxic content.
>
> &nbsp;
>
> ### Utility in an Opaque Ecosystem
> ---
> You are right to point out that the most popular "open-weight" models (like Qwen, DeepSeek, or Kimi) do not release the full training data required for our positional offset-based analysis. However, despite the fact that we cannot conduct the same analysis as in our paper on these models, our work still yields a promising future direction to audit these very models based on our finding: the "sliding window" attack for Membership Inference Attacks (MIAs), as mentioned in our rebuttal.
>
> &nbsp;
>
> ### The Ecosystem Gap and Our Role in Bridging It
> ---
> We recognize the shortcomings you have identified in prevailing “open-weight” practices. Rather than focusing on the limitations of past releases, we believe the most constructive path forward is to help establish a new, more transparent standard. In this spirit, we are pleased to share that our ongoing pretraining of **new 8B and 70B models on 15T permissively-licensed data** only is nearing completion, we will release them with the full transparency we advocate.
>
> Thanks to the robust data indexing capabilities of NVIDIA's Megatron-LM as we confirmed in this work, we will make these models fully open—releasing not just the weights and code, but also the complete training data and the necessary metadata to enable document-level traceability for the community. Our pretraining data for these models explicitly includes the different "Gutenberg FM-probe buckets" across various training stages, to simulate high copyright risk content, allowing for direct, large-scale validation of our findings.
>
> We hope this effort will serve as a contribution that helps inspire a higher standard of transparency—providing the community with the very kind of reference you have rightfully asked for and helping to turn what is technically feasible into common practice.
>
> &nbsp;
>
> ---
>
> Last but not least, in light of this valuable discussion, we will update the Limitations section of our manuscript. We will more explicitly frame the critical gap you identified between our experimental setup and the prevailing practices in the field.
>
> we would like to express our sincere gratitude for your detailed comments and valuable engagement throughout this review process. Your questions, despite being challenging, were instrumental. They pushed us to think more deeply about the practical implications of our work and to clarify its position within the current landscape of LLM development. Thank you once again for your recognition of our work and for helping us improve its clarity and impact.
>
> With sincere appreciation,
>
> The Authors of Submission 12369

---

### Official Review · Reviewer_skq5 · 2025-07-06

**Clarity:** 4
**Significance:** 4
**Originality:** 3
**Rating:** 6
**Confidence:** 3

**Summary:**

This paper focuses on understanding memorization of training data. The authors pretrain three Llama-style transformers on a mix of Project Gutenberg books and web texts, and find (1) short prefixes at the start of the context window lead to highest memorization, whereas shifting this prefix 64-128 tokens deeper can almost eliminate this effect, (2) models that fail to retrieved the memorized sufix will produce nonsensical outputs, and (3) inserting sensitive text further away from the beginning is a simple mitigation strategy.
The authors also show several related secondary analyses related to batch sizing and model size.

**Questions:**

What happens if you move away from greedy sampling?

How does the size of the special anchoring zone change according to model size?

**Ethical Concerns:**

["NO or VERY MINOR ethics concerns only"]

**Limitations:**

Yes

**Quality:**

4

**Strengths And Weaknesses:**

This paper analyzes an important downstream impact of an LLM's tendency to focus on early tokens. The experiments are clear to understand, and the problems and solutions presented are helpful. The authors show results across several model sizes, vary the exposure counts, and ablate batch sizing. The paper also alludes to a connection between memorization and model degeneration, which is helpful.

---

> ### Author Rebuttal · Authors · 2025-07-30
>
> We sincerely thank the reviewer for championing our work and for the thoughtful questions, which helped sharpen our understanding of both the robustness and scalability of positional memorization.
>
> &nbsp;
>
> ### Question 1: What happens if you move away from greedy sampling?
> ---
> Thank you for this important question. To evaluate the robustness of our findings under alternative decoding strategies, we re-ran inference on the repetition-128 FM-Probe bucket using our 8B model with:
>
> - **Beam search** (beam size = 5), reporting the top-ranked output
> - **Nucleus sampling** (top-p = 0.9)
>
> Our core finding—**positional fragility**—remains consistent across all decoding methods. As shown in the table below, `ROUGE-L` scores drop sharply as the prefix is offset deeper into the context, and all methods converge to near-baseline levels (~0.18–0.19) at an offset of 2048 tokens.
>
> &nbsp;
>
>
>
> | Offset | Rouge-L ||| PPL ||| TTR_gen |||
> |--------|---------|-----------|----------|---------|-----------|----------|---------|-----------|----------|
> |        | greedy  | beam  | nucleus  | greedy  | beam  | nucleus  | greedy  | beam  | nucleus  |
> | 0      | 0.965   | 0.999     | 0.877    | 1.030   | 1.011     | 1.164    | 0.538   | 0.541     | 0.536    |
> | 1      | 0.958   | 0.994     | 0.844    | 1.035   | 1.012     | 1.210    | 0.536   | 0.539     | 0.532    |
> | 2      | 0.951   | 0.993     | 0.852    | 1.043   | 1.016     | 1.206    | 0.535   | 0.538     | 0.533    |
> | 4      | 0.913   | 0.978     | 0.778    | 1.073   | 1.021     | 1.329    | 0.530   | 0.533     | 0.530    |
> | 8      | 0.864   | 0.951     | 0.701    | 1.114   | 1.033     | 1.487    | 0.519   | 0.524     | 0.523    |
> | 16     | 0.783   | 0.895     | 0.625    | 1.179   | 1.054     | 1.702    | 0.504   | 0.505     | 0.513    |
> | 32     | 0.652   | 0.800     | 0.495    | 1.298   | 1.090     | 2.123    | 0.474   | 0.469     | 0.503    |
> | 64     | 0.488   | 0.691     | 0.358    | 1.462   | 1.158     | 2.640    | 0.438   | 0.433     | 0.488    |
> | 128    | 0.330   | 0.507     | 0.247    | 1.655   | 1.224     | 3.358    | 0.391   | 0.361     | 0.477    |
> | 256    | 0.255   | 0.354     | 0.203    | 1.773   | 1.282     | 4.022    | 0.346   | 0.297     | 0.466    |
> | 512    | 0.208   | 0.244     | 0.186    | 1.874   | 1.324     | 4.606    | 0.305   | 0.237     | 0.461    |
> | 1024   | 0.197   | 0.202     | 0.186    | 1.906   | 1.352     | 4.930    | 0.284   | 0.206     | 0.458    |
> | 2048   | 0.192   | 0.186     | 0.180    | 1.932   | 1.332     | 5.071    | 0.279   | 0.196     | 0.460    |
>
> &nbsp;
>
> That said, the decoding strategies differ in their behavior:
> - **Beam search**, a deterministic, probability-maximizing strategy, achieves nearly perfect memorization at shallow offsets (e.g., `ROUGE-L` = 0.999 at offset 0), with the lowest perplexity, reflecting high model confidence. However, when recall fails (at deeper offsets), it produces the most severe degeneration—supporting our claim that failed retrieval leads to incoherent outputs.
> - **Nucleus sampling** exhibits lower memorization (lower `ROUGE-L`) at shallow offsets, but much higher lexical diversity at large offsets (e.g., `TTR_gen` = 0.460 at offset 2048), indicating robustness against degeneration. This confirms that sampling-based methods can better mitigate the adverse effects of failed recall.
>
> > We note that, unlike in the main paper where perplexity measures uncertainty over the ground-truth suffix, the `PPL` here reflects uncertainty over self-generated continuations and thus varies with decoding.
>
> &nbsp;
>
> ### Question 2: How does the size of the special anchoring zone change according to model size?
> ---
> We thank the reviewer for this question, which probe the key scaling behaviour of our findings. To investigate, we conducted a fine-grained offset sweep across the first 384 tokens (step size = 16) and applied a `ROUGE-L` threshold of 0.4 to identify the anchoring zone boundary for each model. We found:
> - **1B model**: Anchoring zone ends at offset 16 (ROUGE-L = 0.396)
> - **3B model**: Anchoring zone ends at offset 80 (ROUGE-L = 0.368)
> - **8B model**: Anchoring zone ends at offset 112 (ROUGE-L = 0.376)
>
> &nbsp;
>
> | Offset | Rouge-L (1B / 3B / 8B)     | Ref_PPL (1B / 3B / 8B)     | TTR_gen (1B / 3B / 8B)     |
> |--------|----------------------------|----------------------------|----------------------------|
> | 0      | 0.744 / 0.880 / 0.965      | 1.051 / 1.020 / 1.012      | 0.521 / 0.532 / 0.538      |
> | 16     | 0.396 / 0.762 / 0.783      | 1.095 / 1.038 / 1.074      | 0.485 / 0.516 / 0.504      |
> | 32     | 0.346 / 0.614 / 0.652      | 1.127 / 1.082 / 1.138      | 0.465 / 0.489 / 0.474      |
> | 48     | 0.308 / 0.499 / 0.566      | 1.169 / 1.129 / 1.193      | 0.453 / 0.470 / 0.455      |
> | 64     | 0.282 / 0.447 / 0.488      | 1.203 / 1.175 / 1.237      | 0.447 / 0.451 / 0.438      |
> | 80     | 0.254 / 0.368 / 0.443      | 1.254 / 1.244 / 1.281      | 0.434 / 0.423 / 0.425      |
> | 96     | 0.242 / 0.329 / 0.400      | 1.281 / 1.292 / 1.321      | 0.425 / 0.408 / 0.411      |
> | 112    | 0.228 / 0.303 / 0.376      | 1.323 / 1.318 / 1.374      | 0.417 / 0.405 / 0.398      |
> | 128    | 0.219 / 0.274 / 0.330      | 1.356 / 1.378 / 1.417      | 0.411 / 0.394 / 0.391      |
> | 144    | 0.215 / 0.257 / 0.317      | 1.394 / 1.437 / 1.446      | 0.407 / 0.381 / 0.386      |
> | 160    | 0.210 / 0.241 / 0.296      | 1.428 / 1.485 / 1.472      | 0.405 / 0.367 / 0.374      |
> | 176    | 0.206 / 0.237 / 0.282      | 1.465 / 1.533 / 1.501      | 0.397 / 0.368 / 0.373      |
> | 192    | 0.205 / 0.226 / 0.284      | 1.486 / 1.571 / 1.531      | 0.396 / 0.357 / 0.362      |
> | 208    | 0.204 / 0.218 / 0.265      | 1.520 / 1.598 / 1.563      | 0.394 / 0.358 / 0.358      |
> | 224    | 0.204 / 0.223 / 0.267      | 1.552 / 1.639 / 1.584      | 0.391 / 0.354 / 0.349      |
> | 240    | 0.201 / 0.210 / 0.257      | 1.580 / 1.683 / 1.606      | 0.389 / 0.345 / 0.349      |
> | 256    | 0.200 / 0.214 / 0.255      | 1.609 / 1.723 / 1.634      | 0.383 / 0.343 / 0.346      |
> | 272    | 0.200 / 0.207 / 0.251      | 1.654 / 1.747 / 1.655      | 0.381 / 0.341 / 0.344      |
> | 288    | 0.198 / 0.207 / 0.250      | 1.684 / 1.772 / 1.675      | 0.375 / 0.335 / 0.337      |
> | 304    | 0.198 / 0.209 / 0.239      | 1.724 / 1.811 / 1.697      | 0.378 / 0.335 / 0.337      |
> | 320    | 0.197 / 0.204 / 0.233      | 1.762 / 1.834 / 1.714      | 0.373 / 0.328 / 0.335      |
> | 336    | 0.198 / 0.204 / 0.229      | 1.816 / 1.865 / 1.740      | 0.365 / 0.329 / 0.324      |
> | 352    | 0.196 / 0.205 / 0.229      | 1.832 / 1.888 / 1.757      | 0.365 / 0.326 / 0.329      |
> | 368    | 0.195 / 0.202 / 0.228      | 1.874 / 1.912 / 1.767      | 0.363 / 0.324 / 0.331      |
> | 384    | 0.198 / 0.201 / 0.225      | 1.894 / 1.954 / 1.795      | 0.357 / 0.320 / 0.325      |
>
> &nbsp;
>
> Beyond these thresholds, all models drop toward a plateau `ROUGE-L` of ~0.20. The 8B model converges fully by offset 512 (as seen in Figure 1a), though still slightly above plateau (0.225) at offset 384.
>
> Importantly, this growth is sub-linear. The increase from 1B to 3B (3× params) leads to a 5× expansion of the anchoring zone (16 → 80 tokens), while 3B to 8B (~2.7× params) yields only a 1.4× increase (80 → 112 tokens). Fitting a power law ($\text{Offset} = k \cdot P^a$) to these points gives a scaling exponent of approximately a ≈ 0.34, confirming sub-linear expansion.
>
> Extrapolating this trend, we estimate that a 70B model would require an offset of approximately 1090 tokens for memorization to fully decay to its baseline level. This projection is calibrated using the 8B model’s convergence point at offset 512 and the fitted power law with scaling exponent a ≈ 0.34. We stress that this is a preliminary projection. While the power law provides a reasonable fit for the 1B–8B range, its behavior at larger scales (e.g., 70B) remains unverified and must be tested empirically.
>
> This finding also has practical implications. While offsets in the 1K-token range may still be manageable in modern long-context settings, they represent a nontrivial memory region that remains vulnerable to leakage. For use cases demanding high confidentiality, relying on positional mitigation alone may be insufficient.
>
> To that end, our ongoing pretraining of a new **8B** and **70B** model on **15T** tokens is designed to explore these dynamics at scale. In this new run, we are using the Gutenberg FM-Probe buckets to precisely measure how pretraining-time mitigation techniques, such as Goldfish Loss [1], interact with positional fragility. This large-scale pretraining run is nearing completion, and a full analysis of these effects will be presented in future work.
>
> &nbsp;
>
> ---
>
> Finally, to ensure full transparency and reproducibility, we will update the appendix of our manuscript to include the complete results from the additional experiments conducted for this rebuttal. We hope these additions resolve the raised uncertainties and make verification straightforward.
>
> &nbsp;
>
> **References**:
> [1] Abhimanyu Hans, et al., "Be like a Goldfish, Don't Memorize! Mitigating Memorization in Generative LLMs," Advances in Neural Information Processing Systems, vol. 37, pp. 24022-24045, 2025.

---

> > ### Comment · Reviewer_skq5 · 2025-08-06
> >
> > Thank you for the follow-up analysis. I am confirming my support for publication and my score.

---

> ### Author Response · Authors · 2025-08-07
> **Thank You**
>
> Dear Reviewer skq5,
>
> We are writing to express our sincere gratitude for your exceptionally strong support for our paper. Your insightful questions prompted valuable new analysis that we believe have meaningfully improved the manuscript. We deeply appreciate you taking the time to read our rebuttal and confirm your unwavering support and score.
>
> With sincere appreciation,
>
> The Authors of Submission 12369

---

### Note · Authors · 2025-08-13

Dear AC, SAC and PC,

We sincerely thank all reviewers for their constructive feedback. We're pleased they recognized our work's key strengths, including the novelty of our "positional fragility" finding, the rigor of our experiments, the link to text degeneration, and our effective mitigation strategy.

The feedback highlighted two main areas, which we addressed with new work:

* **Robustness**: To address robustness concerns, we conducted a new continued pretraining experiment on a Llama 3.2 1B model, analyzed different decoding strategies, and ran a 13-gram contamination analysis to rule out copying. All results validated our findings.

* **Practicality**: We clarified that our mitigation is a Proof of Concept that doesn't harm downstream performance. We also demonstrated that document-level traceability is technically feasible in modern frameworks like Megatron-LM and shared our commitment to bridge this gap with new, fully transparent 8B and 70B models.

&nbsp;

The discussions led to positive outcomes: Reviewer skq5 confirmed their 'Strong Accept'; Reviewer mkAN confirmed our new experiments "addressed most of my concerns"; the in-depth discussion with Reviewer vjB5 was invaluable; and while we didn't hear back directly from Reviewer Mw7L, they raised their scores, so we assume their concerns were resolved.

&nbsp;

Our Commitment to Revisions
1.  **Updating the Appendix** with full results from all new experiments and analyses.
2.  **Revising Key Statements** to better frame our mitigation as a Proof of Concept.
3.  **Expanding the Limitations section** to address the gap with current practices.
4.  **Adding contamination ratio between Gutenberg and FineWeb-Edu** to demonstrate experimental cleanliness.

&nbsp;

We are confident this process has made our contribution significantly stronger and we hope it now meets the high standards for publication at NeurIPS.

---

### Decision · Program_Chairs · 2025-09-17

**Decision:**

Accept (poster)

**Comment:**

The paper explorers the relationship between memorization of token sequences an LM sees during training and the offset of said token sequence within the batch. The paper finds that token sequences seen at the start of the batch are more likely to be memorized. In addition, in cases where the LM fails to produce the true continuation to a prompt known to be in the training dataset, the LM is also more likely to exhibit degeneration, outputting an incoherent continuation.

Reviewers had very mixed feelings about this paper with scores ranging from "strong accept" to "borderline reject."

The positive reviews note that paper's findings are truly novel, analyzing a facet of memorization which has not been analyzed before. Multiple reviewers appreciated the comprehensive experiments. Reviewers also liked that the paper didn't just include an analysis of the phenomenon, but proactively suggested a way to make use of as a mitigation strategy against undesired memorization.

One criticism of the paper was that the proposed mitigation strategy is unrealistic, as it requires model trainers to know a priori which parts of their training data are sensitive. There are also concerns the paper does not do a job enough job explaining its results that are counterintuitive relative to prior work (such as the result that smaller batch sizes are better for reducing memorization). Another criticism is that the paper doesn't clearly the settings used for its definitions of things like the offset. Though not mentioned by the reviewers, I find the distinction between "context window size" and "batch size" is also not made clear. Are these always the same?

Overall, I think the paper presents a novel analysis that is a valuable contribution to the study of memorization in language models. I recommend its acceptance.

A couple other comments from my read of the paper (though not mentioned by any of the reviews):

I strongly recommend the authors revisit their figures in order to ensure all font sizes are legible. The text in Figure 4 is completely illegible when the paper is printed. You may consider making your table text \small sized in order to make room for a larger Figure 4. The x and y axis labels in your other figures could also benefit from big font sizes.

The phrase "prior work implicitly assumed uniformity by probing only from the beginning of training sequences" seems like a misrepresentation of prior work. Prior work such as Carlini et al (2023) prompted with the starts of documents, but the starts of documents are not necessarily the starts of training sequences, since many modern LLM are trained with packing, where multiple documents are packed into a training sequence (and also, particularly long documents are broken into multiple training sequences). I believe the abstract should be revised, and this distinction should be addressed in the text.